



# 1   Catchment power and the joint distribution of elevation
# 2   and travel distance to the outlet.

Leonard S. Sklar[1], Clifford S. Riebe[2], Claire E. Lukens[2], Dino Bellugi[3]
[1]Department of Earth & Climate Sciences, San Francisco State University, San Francisco, CA 94132 USA
[2]Department of Geology and Geophysics, University of Wyoming, Laramie, WY 82071 USA
[3]Department of Earth, Atmospheric and Planetary Sciences, MIT, Cambridge, MA 02139 USA
Correspondence to L.S. Sklar (leonard@sfsu.edu)
**Abstract**  The delivery of water, sediment and solutes by catchments is influenced by the distribution of
source elevations and their travel distances to the outlet. For example, elevation affects the magnitude and
phase of precipitation, as well as the climatic factors that govern rock weathering, which influence the
production rate and initial particle size of sediments. Travel distance, in turn, affects the timing of flood
peaks at the outlet and the degree of sediment size reduction by wear, which affect particle size
distributions at the outlet. The distributions of elevation and travel distance have been studied extensively
but separately, as the hypsometric curve and width function. Yet a catchment can be considered as a
collection of points, each with paired values of elevation and travel distance. For every point, the ratio of
elevation to travel distance defines the mean slope for transport of mass to the outlet.  Recognizing that
mean slope is proportional to the average rate of loss of potential energy by water and sediment during
transport to the outlet, we use the joint distribution of elevation and travel distance to define two new
metrics for catchment geometry: "source-area power," and the corresponding catchment-wide integral
"catchment power." We explore patterns in source-area and catchment power across three study catchments
spanning a range of relief and drainage area. We then develop an empirical algorithm for generating
synthetic source-area power distributions, which can be parameterized with data from natural catchments,
and used to explore the effects of topography on the distribution on fluxes of water, sediment, isotopes and
other landscape products passing through catchment outlets. This new way of quantifying the three-
dimensional geometry of catchments may provide a fresh perspective on problems of both practical and
theoretical interest.

## 27   1. Introduction

The physical and ecological dynamics of rivers are influenced by upstream sources of water,
solutes, and sediment. These materials are produced at rates that vary from source to source depending on
factors such as precipitation, weathering, erosion, and ecosystem productivity. Spatial variations in these
factors commonly correspond to differences in elevation. For example, elevation influences both the
magnitude and phase of precipitation (Roe, 2005; Minder et al., 2011), the climatic factors that govern rock
weathering (White and Blum, 1995; Riebe et al., 2004), the particle size and production rate of sediment
from slopes (Marshall and Sklar, 2012; Riebe et al., 2015), and both the distribution of biomes (Lomolino,



2001) and their net primary productivity (Raich et al., 1997). Thus elevation is a fundamental characteristic
of the source areas that supply water, solutes, and sediment to catchment outlets.

Along the journey from source to outlet, material is mixed together with products of other sources

and altered by chemical, physical, and biological processes. The mixing and alteration of materials depends
in part on the travel distance between the source and outlet. For example, travel distance influences the
generation of flood waves (Richie et al., 1989), the liberation of solutes and nutrients from soil and
sediment (Gaillardet et al., 1999; Jin et al., 2010), the physical breakdown of sediment in streams (Attal and
Lave, 2006), and the decomposition of organic matter (Taylor and Chauvet, 2014). Thus travel distance is
another fundamental aspect of the link between source and outlet for water, solutes, sediment, and
nutrients.

Together, the effects of elevation and travel distance should govern the amount, timing, and

composition of fluxes from catchments. However, previous work has explored the distributions of elevation
and travel distance separately, without consideration of their joint distribution. The distribution of
elevations – known as hypsometry – reveals the vertical structure of a catchment and has been used to
quantify landscape development, identify geomorphic process regimes, and understand the sensitivity of
land area to changes in sea level (Strahler, 1952; Lifton and Chase, 1992; Brozovic et al., 1997;
Brocklehurst and Whipple, 2004; Algeo and Seslavinsky, 1995). Meanwhile, the distribution of travel
distances – known as the width or area function – reveals the horizontal structure of catchments and has
been used to characterize catchment shape, identify channel branching structure, and understand
hydrographs (Gupta and Mesa, 1988; Rinaldo et al., 1995; Sklar et al., 2006; Moussa, 2008; Rigon et al.,

2015).

Although both the hypsometry and width functions of catchments have been widely studied, to our

knowledge elevation and travel distance have only been considered together in an analysis of the
hypsometry of channel network links (Gupta and Waymire, 1989) and in plots of longitudinal profiles of
trunk streams and tributaries (Rigon et al., 1994). Thus, previous research has overlooked the insights that
might be gained by analyzing hillslopes and channels together as a collection of paired values of elevation
and travel distance. Some questions that might be addressed by such an analysis include: Which if any
aspects of the joint distribution of elevation and travel distance are common from one catchment to the
next? What are the most revealing measures of differences in the distributions across different catchments?
Do the distributions differ in ways that systematically reflect the factors that drive landscape evolution,
such as weathering, climate, and tectonics?

Here we address these questions using topographic data from three catchments of differing area

and relief. First we explore how the distributions of elevation and travel distance vary across our study
catchments. Then we show how elevation and travel distance can be combined into a single quantity,
referred to here as catchment power because it expresses the rate of potential energy dissipation of water
and sediment as they travel down slopes. Next, using our analyses of the elevation and travel distance
distributions from the study catchments, we develop an approach for generating synthetic catchments that



capture many features of power distributions in natural landscapes and thus can be used to explore how
factors such as area, relief, and profile concavity influence catchment power. Finally, we discuss how our
approach provides a new framework for understanding how rivers are influenced by upstream sources of
water, solutes, and sediment in catchments.

## 2. Elevation and travel distance in natural landscapes

To explore how joint distributions of elevation and travel distance vary in natural landscapes, we
chose catchments drained by Inyo Creek, Providence Creek, and the Noyo River, all in California, USA
(Fig. 1). Each of these catchments has been featured in previous studies of the production and delivery of
water, solutes, and sediment from slopes to channels. Thus our selection of sites allows us to link analyses
of elevation and travel distance distributions to existing research on physical, chemical, and biological
processes in the catchments. All of the catchments are developed in mountain landscapes, where the
products of runoff, weathering, and erosion reach the outlet without any long-term interception in
floodplains or lakes; thus, the travel distance distributions should strongly reflect transport processes in the
catchments. At each site, we extracted elevations from a 10-m digital elevation model (DEM) and
calculated travel distance to the outlet using a steepest descent algorithm (Tarbotton, 1997). The
catchments span a range in relief, drainage area, and mean slope (Table 1), and thus also a range in the
populations of paired values of elevation and travel distance (Fig. 1).

### 2.1    Study sites

The Inyo Creek catchment spans 2 km of relief over 4 km of travel distance on the eastern slope of
the High Sierra (Table 1). Unlike some of its neighboring catchments along the range, it has never been
scoured by glaciers, making it ideal for comparison of sediment production and landscape evolution in
glaciated and non-glaciated terrain (Riebe et al., 2015; Stock et al., 2006; Brocklehurst and Whipple, 2002).
Moreover, the catchment spans a range in the relative importance of physical, chemical, and biological
weathering from its warm, gently sloped, low elevations to its cold, steep headwaters.
On the other side of the Sierra Nevada, Providence Creek  spans 1 km of relief over 8 km of travel
distance (Table 1). This catchment is part of the Southern Sierra Critical Zone Observatory, which has been
the focus of numerous recent studies of hydrology, biogeochemistry, and geomorphology (e.g., Bales et al.,
2011; Hunsaker and Neary, 2012; Hunsaker et al., 2012; Goulden and Bales, 2014; Holbrook et al., 2014;
Hahm et al., 2014). Precipitation in the upper half of the catchment dominantly falls as snow, whereas
precipitation in the lower half dominantly falls as rain. Unlike the roughly continuous concave ridge and
channel profiles of Inyo Creek, catchment topography in Providence Creek exhibits a pronounced step in
elevation of both the channel and ridge profiles (Fig. 1). Steps like these, which are common on the
southwestern slope of the Sierra Nevada, have been interpreted to reflect a feedback between weathering
and erosion (Wahrhaftig, 1965).





Farther to the northwest, in the California Coast Ranges, the Noyo River catchment spans 0.9 km
of relief over 20 km of travel distance. Thus the catchment is significantly larger and more gently sloped on
average than either of the other two study catchments. The catchment has a long history of intensive timber
harvests and has been the site of numerous studies of the effects of land use on in-stream habitat (Burns,
1972; Lisle, 1982; Leithold et al., 2006; ) and the role of topography and channel network structure in the
production and delivery of sediment from slopes to channels (Dai et al., 2004; Sklar et al., 2006).
**2.2 Spatial distributions of elevation and travel distance**
The maps in Figure 2 show the spatial distributions of elevation and travel distance across each
catchment. Broadly, travel distance and elevation covary in space; the highest elevations in each catchment
tend to be further away from the outlet. However, in detail, elevation contours are not aligned with contours
of equal travel distance; in general the elevation contours exhibit higher planform curvature than travel
distance contours. Thus, for a given elevation contour, travel distances are longest in the valley axis and
shortest at the ridges. Conversely, for a given travel distance, elevations are highest at the ridges and lowest
in the valley axis. These patterns are especially clear at Inyo Creek (Fig. 2a) and Providence Creek (Fig.
2b), which drain small, relatively undissected catchments.
The patterns in elevation and travel distance in the Noyo River catchment are more complex (Fig.
2c), in part because it is more deeply incised by multiple high-order trunk streams. At ridges that separate
these trunk streams, travel distance can vary considerably from one side of the ridge to the other. Thus
nearby points that share the same elevation can have very different travel distances. For example, along the
central ridge, which runs along the catchment's axis, points on the south side of the ridge drain to a more
sinuous and thus longer southern trunk stream, giving them longer travel distances to the outlet than points
on the northern side. For the same travel distance, points occur at higher elevations in the northern, less
sinuous trunk stream.
**2.3 Hypsometry and the width function**
The spatial patterns shown in the maps are reflected in both the hypsometry and the width
function, which are the conventional ways of displaying distributions of elevation and travel distance
separately (Fig. 3). For example, hypsometry shows that most of the Inyo Creek catchment occurs at mid
elevations (Fig. 3a), because the catchment narrows both at low elevation near the outlet and at high
elevation near the catchment divide (Fig. 2a). This differs from the hypsometry of Providence Creek, where
most of the catchment area occurs at higher elevations, above the pronounced step in the topography.
Meanwhile, at the Noyo River site, the majority of area occurs at lower elevations, because the catchment
is deeply dissected, with wide valley bottoms and steep, narrow ridges.
Hypsometry reveals differences in the vertical structure of the catchments, whereas the width
function reveals differences in planform structure, which are governed in part by differences in the shapes





of the catchment boundaries. For example, the distribution of travel distances at Inyo Creek is symmetrical,
reflecting the roughly oval shape of the catchment. Meanwhile, at Providence Creek, the distribution of
travel distances is bimodal, reflecting the narrowing near the middle of the catchment. At the Noyo River
site, the travel-distance distribution is skewed, with the majority of the area at long travel distances,
reflecting the widening of the catchment with increasing distance from the outlet that is evident in Figure
2c.
**2.4 Joint distributions of elevation and travel distance**
Figure 3 shows that much can be learned from the distributions of elevation and travel distance
plotted alone. However, they do not reveal information contained in the distribution of paired values of
elevation and travel distance. One particularly insightful index that can be missed is the ratio of elevation to
travel distance, which is the mean slope for water, solutes, and sediment on a path of steepest descent from
source to outlet. The ranges in elevations and travel distances from these three catchments imply that the
distribution of mean slopes differ markedly across our sites (Table 1; Fig. 1). These differences likely
correspond to differences in factors such as water-transit times, sediment breakdown rates, and channel
morphology. Although information on the distribution of mean slopes is embedded in both the hypsometry
and the width function, it cannot be extracted from either of them plotted alone or even plotted side by side
(Fig. 3).
To overcome the limitations of separate plots of vertical and horizontal structure, we plotted the
joint distribution of elevation and travel distance for every point in each of the catchments in Figure 4.
These plots show both the long profile of the channel network and the distribution of hillslope sources,
which account for more than 98% of the source area in each catchment. A number of similarities emerge
across the sites (Fig. 4a-c). Strikingly, at the highest elevations for any given travel distance, sources are
aligned in steeply-sloped tendrils of data that coalesce at lower elevations. These tendrils represent hillslope
sources aligned along common flow paths that cluster together into narrow groups. Equally striking are the
gaps between the tendrils, which represent paired values of elevation and travel distance that do not occur
anywhere in the catchment. Meanwhile, some paired values are so common that they overlap, particularly
along flowpaths that converge near the mainstem channel. Thus the joint distribution plots generally show
dense concentrations of data points at low elevations for any given travel distance.
Bivariate frequency distributions help shed light on the degree of clustering and overlap of data at
shared values (Fig. 4 d-f). These binned representations of the raw data show that, for a given travel
distance, as elevation decreases, data point density generally increases to a peak and then quickly tapers to
zero. They also show that the density of paired values is highest at 60 and 80% of the maximum travel
distance, with a tapering in point density at both the upstream and downstream ends of the catchment.
Although the joint distributions are similar in some respects across the catchments, they also
exhibit significant differences that cannot be inferred from the conventional representations of vertical and
horizontal catchment structure in Fig. 3. For example, the relative slopes of the tendrils and the channels





differ markedly. The tendrils are much steeper than the mainstem channel profile in the Noyo River
catchment (Fig. 4f). Conversely, in the other two catchments, the tendrils and the main channel profile have
similar slopes, especially at Providence Creek. These differences likely arise at least in part due to the
difference in scale of the watersheds; in the Noyo River catchment, some of the individual tendrils
encompass large areas, similar in scale to the entire Inyo and Providence Creek catchments. Thus we
interpret the tendrils along the Noyo River to be tributary catchments that are similar to the Inyo and
Providence Creek catchments, with tendrils of their own that are only slightly steeper than the local
tributary channel slopes.
Perhaps the most striking difference among the catchments can be seen in the distributions of
mean slope along the travel path to the outlet, which we calculate as the ratio of the paired values of
elevation and travel distance (Fig. 5a-c insets). Swaths of common mean slope appear as linear trends
through the joint distributions of elevation and travel distance (Fig. 5a-c), or as contours on a planform
view of the catchment (Fig. 5d-f). In each catchment the contours of mean slope (Fig. 5d-f) differ markedly
from the contours of elevation and travel distance (Fig. 2). Mean slopes are relatively steep and span a
relatively narrow range at Inyo Creek (Fig. 5c) compared to the Noyo River catchment (Fig. 5f).
Providence Creek is distinguished by a peak in mean slopes (Fig. 5b) corresponding to the upper half of
catchment, above the step in the topography (Fig. 5e).
Mean slope quantifies the ratio between elevation and travel distance, and thus is a single metric
that combines two fundamental attributes of source areas in catchments. The distributions of source
elevation, travel distance, and thus mean slope are ultimately set by the erosion and transport processes that
produce and deliver sediment from slopes to channels. Thus spatial variations in mean slope, such as those
shown in Fig. 5, may be closely linked to spatial variations in the production and delivery of water, solutes,
and sediment.
**3 Source-area and catchment power**
To develop a mechanistic framework for linking distributions of source-area mean slope with
catchment processes, we introduce the concept of source-area power, which integrates elevation, travel
distance, and the production rate of material on slopes. In the derivation that follows, we consider a mass
($M$) of transportable material (such as water, solutes, or sediment) produced at a source elevation $z$ on a
hillslope and delivered downstream to an elevation $z_o$ at the catchment outlet. The potential energy ($E$) of
the material at the source, relative to the outlet is given by Equation 1:
$$E_{i,j} = M_{i,j}gR_i = \rho_{i,j}A_ih_{i,j}g\left(z_i - z_o\right) \tag{1}.$$

Here $g$ is acceleration due to gravity, $R$ is relief (i.e., the difference in elevation between the source and
outlet), $\rho$ is density, $h$ is the thickness of the material produced at the source, $A$ is the area of the source
(one pixel in a DEM), the subscript $i$ refers to the specific source location on the slope, and the subscript $j$





refers to the type of material (i.e., water, solutes, or sediment). In the case of solutes, $h$ refers to the
equivalent thickness of chemical erosion needed to account for the mass loss due to production of solutes.
At each source, potential energy is produced at a rate ($\Omega$) that is proportional to the production
rate ($Q$) or flux of material from the source, as shown in Equation 2:

$$\Omega_{i,j} = Q_{i,j} g R_i = \rho_{i,j} A_i \frac{\partial h_{i,j}}{\partial t} g(z_i - z_o) \tag{2}$$

Here, the definition of $\partial h / \partial t$ (in dimensions of length per time) depends on the process considered. For
water produced by precipitation, $\partial h / \partial t$ is the precipitation rate. For sediment produced by erosion, $\partial h / \partial t$ is
the physical erosion rate. For solutes produced by chemical erosion, $\partial h / \partial t$ is the equivalent to the chemical
erosion rate. In all cases, $\Omega$ has dimensions of power.
On its journey to the outlet, the material loses its potential energy. This energy is converted to
kinetic energy and is primarily lost to heat due to friction. In the case of sediment, some of the energy is
consumed when particles are abraded and shattered during collisions with other particles and the channel
bed. Thus it may be useful in the context of geomorphic work to think of the power expended by the water
or sediment over the travel distance ($L$) between the source and outlet, as shown in Equation 3:

$$\omega_{i,j} = \frac{Q_{i,j} g R_i}{L_i} = \rho_{i,j} A_i \frac{\partial h_{i,j}}{\partial t} g \frac{(z_i - z_o)}{L_i} \tag{3}$$

Here $\omega$ is the source-area power, which has dimensions of power per length, and $(z_i - z_o)/L_i$ is the mean slope
along the travel path from the source to outlet. The concept of source-area power allows us to explore the
possible implications of variability in the ratio of elevation to travel distance (i.e., the mean slope) on the
production and delivery of water, solutes, and sediment across catchments.
For example, in landscapes where the rate of precipitation or erosion is spatially uniform, we
expect the distribution of source-area power for the water or sediment to be identical to the distribution of
the mean slopes of source areas. In contrast, in landscapes where rates of precipitation and erosion are
spatially variable and sometimes correlated (Reiners et al., 2003;, Burbank et al. 2003), we expect the
distributions of power and mean slopes to differ. This is the case at Inyo Creek where mean annual
precipitation increases with elevation from 290 mm yr[-1] at the outlet to 710 mm yr[-1] at the catchment divide
(Prism Climate Group, 2014), and the rate of production of sediment by erosion has been estimated to
increase exponentially with elevation from 0.03 mm yr[-1] at the outlet to 1.5 mm yr[-1] at the divide (Riebe et
al., 2015). When we combine these relationships for water and sediment production with the distribution of
mean slopes using Equation 3, we arrive at maps showing the spatial distributions of source-area power for
the two materials, water and sediment (Fig. 6a-b). In both cases, the power contours are stretched towards
the catchment divide, relative to the case of uniform precipitation and erosion (equivalent to Fig. 5a),
especially in the case of spatially varying erosion (Fig. 6b), due to the nonlinear relationship between
erosion rate and elevation.





Because the altitudinal gradients in erosion and precipitation are known, we can use them to

explore how the source-area power of water varies across the catchment, relative to the amount of sediment
that must be produced on hillslopes and transported to the outlet, assuming steady state. We define a
dimensionless ratio ( $\omega_{w,s}^{*}$ ) that quantifies the source-area power of water per mass of sediment eroded at
an individual pixel, $i$:
$$\omega_{w,s}^{*} = \frac{\omega_{i,w}}{gQ_{i,s}} = \frac{\rho_w \left( \partial h_{i,w}/\partial t \right)}{\rho_s \left( \partial h_{i,s}/\partial t \right)} \frac{\left( z_i - z_o \right)}{L_i} \tag{4}$$

Here the subscript $w$ refers to water produced from precipitation, and the subscript $s$ refers to sediment
produced from erosion. The spatial distribution of $\omega_{w,s}^{*}$ shows that the relative amount of water power
available to produce and transport sediment increases from 36 to 653 (mean ± standard deviation =
254±149) from the headwaters to the catchment mouth (Fig. 6C). We interpret this factor of 18 change to
reflect shifts from headwaters to outlet in dominant geomorphic processes. For example, on headwater
slopes where less water is available and $\omega_{w,s}^{*}$ is lowest, we might expect that sediment transport is
dominated by gravity-driven mass wasting and that weathering is dominated by physical rather than
chemical processes. In contrast, on slopes near the catchment mouth, where $\omega_{w,s}^{*}$ is highest, we might
expect that sediment transport is dominated by water-driven erosion (e.g., via sheetwash and channelized
flow), and that weathering is dominated by chemical processes. This is broadly consistent with field
observations: headwater slopes consist of steep, landslide-dominated bare bedrock, whereas slopes near the
catchment outlet are gentler, more vegetated, and soil mantled, implying that chemical weathering is
favored by longer residence times of water and sediment (Riebe et al., 2015).

To characterize power at the scale of whole catchments, we sum Equation 3 over the entire

contributing area, using Equation 5
$$\omega_{c,j} = g \sum_{i=1}^{i=N} \rho_{i,j} A_i \frac{\partial h_{i,j}}{\partial t} \frac{\left( z_i - z_o \right)}{L_i} \tag{5}.$$

Here $\omega_{c,j}$ is the catchment-integrated source-area power for the material of interest $j$, or, more simply,
"catchment power." It expresses the total power expended as the potential energy of material produced
throughout the catchment is lost along flow paths to the outlet. For Inyo Creek, the total catchment power
for water is 166 W m$^{-1}$, while the total catchment power for sediment is 0.122 W m$^{-1}$. The ratio of
catchment power for water to sediment is 136. This ratio reflects the combined effects of the steep
altitudinal increase in erosion rates, the more modest altitudinal increase in precipitation rates, and how
these trends map into the joint distribution of elevation and travel distance.

New theory and data from other landscapes are needed to interpret spatial variations in power

across individual catchments and to understand why they vary from catchment to catchment. For example,



we might expect to find a different spatial distribution of water-sediment power ratios, relative to Inyo
Creek, in a catchment with a different hypsometry and width function. Likewise, the spatial distribution of
source-area power would differ greatly in a catchment responding to accelerated base-level lowering, with
faster erosion rates near the outlet. Moreover, we might expect the ratio of water to sediment catchment
power to vary considerably from catchment to catchment across gradients in climate and tectonics.
Understanding these variations could provide fresh insights into the geomorphic processes that shape
landscapes.

Although our analysis of power at Inyo Creek focused on the production of water and sediment, it

can be extended to any material that varies in production rate with altitude or varies in delivery to the outlet
as a function of travel distance. For example, production rates of solutes, nutrients, contaminants, and even
cosmogenic nuclides could be substituted for the production rate terms in Equations 2-5. Thus it should be
possible to use the new frameworks of source-area and catchment power to model, and thus better
understand, both the spatial distribution and catchment-integrated effects of geomorphic, geochemical, and
ecosystem processes.

Our analysis of Inyo Creek shows how the power framework can be applied to natural landscapes

using a DEM. However, factors, such as climate, topography, and tectonics, which might influence power
and thus merit further investigation, are closely coupled together. This makes it difficult to isolate any
single factor of interest in comparisons of power across catchments. Moreover, some catchments, such as
Providence Creek, have peculiarities in shape and structure that dominate patterns of power (Fig. 5b) and
thus might confound comparisons of one catchment to the next. To overcome the limitations of using
DEMs from individual catchments, we developed an approach that generates synthetic catchments based on
scaling relationships for catchment geometry and topography. Thus we can systematically explore how
variations in factors such as area, relief, and profile concavity influence the distribution of source-area and
catchment power in landscapes. In the next section we show that our synthetic catchments capture the
fundamental characteristics of the joint distribution of elevation and travel distance in landscapes. Thus we
can use them to isolate and thus study the influence of physical, chemical and biological factors that govern
catchment processes.
**4 Synthetic joint distributions of elevation and travel distance**

Our goal in developing synthetic catchments is to generate realistic joint distributions of elevation

and travel distance (e.g., that are comparable to those shown in Fig. 3). Equations 3-5 show that this should
be sufficient to quantify distributions of source-area and catchment power.  Hence there is no need for a
spatially explicit representation of topography, because calculating source-area power does not require
information about spatial position of channels or topographic factors such as hillslope gradient or curvature.
Populating the joint distribution of elevation and travel distance only requires specifying the upper and
lower boundaries at each travel distance and then distributing area across elevations in the space between
the boundaries. Although theory is available to generate main-stem longitudinal profiles that could serve as



a realistic lower boundary of the distribution, we are unaware of any theory for predicting ridge profiles
and thus delineating a realistic upper boundary. Most importantly, to our knowledge, no theory is available
for populating the elevation distribution for a given travel distance between the upper and lower
boundaries, without creating a spatially explicit synthetic DEM using a landscape evolution model
(Coulthard, 2001; Willgoose, 2005; Tucker and Hancock, 2010).

As a starting point for overcoming these limitations, we adopt a statistical, empirical approach,

using Inyo Creek as a prototype for a relatively simple, symmetrical low-order catchment. We start with the
actual maximum and minimum elevations at each travel distance and use a statistical optimization
procedure to find the best-fit distribution of elevations. We then develop expressions for the upper and
lower boundaries at each travel distance and use the best-fit area-versus-elevation function to define a fully
synthetic joint distribution of elevation and travel distance.
**4.1 Area-versus-elevation at each travel distance**

To find the best-fit relationship between area and elevation at each travel distance, we parsed the

Inyo Creek catchment into forty-seven 100-m wide travel distance bins (Fig. 7A). Figure 7B shows
distributions of area with elevation for seven representative travel distance bins. Inspection of figure 7B
suggests that the area under the curves scales with local relief (i.e., the width across the base of the curve),
and that the distributions are consistently right skewed, with more area at the lower elevations. When we
sum area and relief across all bins, and plot the fractional area versus fractional relief for each bin, we find
that the data roughly follow a 1:1 line (Fig. 7C). We obtain a similar result for a variety of bin spacings,
which suggests that the area-elevation relationship is self similar: when the upper and lower boundaries are
farther apart (i.e., when local relief is higher), the area contained within the travel distance bin increases in
direct proportion to the difference in relief. This permits a collapse of the distributions of elevation for each
travel distance bin, by normalizing elevation with local relief, and area by total area in the bin.  Figure 7D
shows the normalized hypsometry for travel distance bins spanning the entire Inyo Creek catchment.  The
broad consistency of the shapes of the normalized distributions suggests that a single functional form could
represent the central tendency, spread and even the skew of the distribution of area with elevation for any
travel distance across the catchment.

The beta distribution has a simple functional form that captures two key characteristics of the

normalized area-elevation relationships: it is bounded by 0 and 1, and it can have right-skew depending on
the values of its two shape factors, $\alpha$ and $\beta$. Thus a beta distribution is well suited to generating synthetic
distributions of area as a function of elevation.

A generic form of the beta distribution is shown in Equation 6

$$f_\beta = x^{\alpha-1}\left(1-x\right)^{\beta-1}$$

(6).





Here $f_\beta$ is the height of the beta distribution at point x, where x ranges from 0 to 1 and the sum of area under
the curve is equal to 1.

To find the values of $\alpha$ and $\beta$ that correspond to the best fit between the area-elevation data and

the beta distribution across all travel distances at Inyo Creek, we first converted Equation 6 to Equation 7
for dimensional consistency.
$$f_{A(z,L)} = A_L \left(z^*\right)^{\alpha-1} \left(1-z^*\right)^{\beta-1} \tag{7}.$$

Here, $f_{A(z,L)}$ is the height of the scaled beta distribution at elevation z in travel distance bin L, $A_L$ is the area
in the travel distance bin, and $z^* = \left(z - z_C\right)/\left(z_R - z_C\right)$ where $z_C$ is the elevation of the channel, and $z_R$ is
the elevation of the ridge.

By applying Equation 7 to each travel distance bin, we can generate a synthetic joint distribution

of elevation and travel distance. We then can calculate the misfit between the synthetic and actual joint
distributions as the square root of the mean squared differences (RMSE) at each elevation and travel
distance. To find the best-fit parameters, we used an optimization algorithm to search for the pair of shape
factors that minimize the misfit. For Inyo Creek data, with 100 m travel distance bins, and 40 m elevation
bins (Fig. 7), the best-fit $\alpha$ is 2.6 and best-fit $\beta$ is 3.4. The objective function for this case is shown in
Figure 8. The best-fit parameters yield a beta distribution that follows the trend in the normalized area
distributions shown in Figure 7D.

To quantify the model performance, we use the Nash-Sutcliffe model efficiency statistic (*NS*)

(Nash and Sutcliffe, 1970), which is calculated as
$$NS = 1 - \frac{\sum \left(f_{A-Model} - f_{A-Data}\right)^2}{\sum \left(f_{A-Mean} - f_{A-Data}\right)^2} \tag{8}.$$

Here the subscript 'model' refers to the predictions of Equation 7, 'data' refers to the DEM, and 'mean'
represents a uniform area density in each bin equal to the total area divided by the number of distance and
elevation bins containing data. A model efficiency of 1 implies a perfect match between predictions and
observations. An efficiency of 0 indicates that model predictions are only as accurate as simply using the
mean of the observed data. Less than zero efficiency (NS < 0) implies that the observed mean is a better
predictor than the model. In other words, the closer the model efficiency is to 1, the more accurate the
model is. For this particular binning scheme (100 m distance and 40 m elevation bins), the Nash-Sutcliffe
model efficiency statistic for Inyo Creek is 0.41, indicating good but not excellent agreement with the
topographic data.

To explore the sensitivity of model performance to spatial resolution of the binning scheme, we

repeated the optimization procedure described above for a range of travel distance and elevation bin sizes.
As shown in Figure 9A, the NS values are generally higher for larger bin sizes (i.e. fewer bins), reaching a
local maximum (NS > 0.7) for 400 m travel distance bins. Model efficiency approaches 1.0 (NS > 0.9) for a





single distance bin, which is equivalent to fitting the whole catchment hypsometry with a single beta
distribution curve. The best-fit values of the beta distribution shape parameters vary considerably with the
size of the distance and elevation bins, and are highly correlated (Fig. 9B); the range of resulting
distribution shapes are illustrated in Figure 9C.
These results reveal a tradeoff between model performance and spatial resolution. They also
suggest that, to first order, Equation 7 can capture much of the structure of area as a function of relief at
Inyo Creek. To the extent that we can think of Inyo Creek as a prototypical catchment, we can use Equation
7 to generate synthetic joint distributions of elevation and travel distance for other catchments, with
different channel and ridge profiles.
The good fit between the modeled and observed joint distributions of elevation and travel distance
at Inyo Creek arises in part because the actual profiles of the channel and ridge were used as envelopes on
the area-elevation distributions. This ensures that the boundaries of the modeled joint distribution
correspond to actual topographic data. To generate a fully-synthetic joint distribution of elevation and
travel distance, an approach is needed that not only distributes area across elevations but also produces
synthetic channel and ridge profiles that define the upper and lower boundaries of elevation as a function of
travel distance.
**4.2 Main-stem channel and ridge profiles**
For any travel distance, the lowest elevation will be on the channel main-stem. Thus, the main-
stem long profile is the lower boundary for the joint distribution of elevation and travel distance. Channel
elevations ($z_C$) are commonly modeled as a power function of travel distance ($x$) along the main stem from
the outlet to the upstream limit of fluvial processes (i.e., the distance to the "channel head", denoted $x_{ch}$). As
elaborated in the appendix, here we derive an expression for channel elevation that extends all the way to
the top of the catchment, at the point where the valley axis meets the drainage divide.
From the outlet to $x_{ch}$, the elevation of the channel can be written as:
$$z_C = k_C \left[ \left( L_{\max} \right)^{1-\theta H} - \left( L_{\max} - x \right)^{1-\theta H} \right] \text{for } 0 \leq x \leq x_{ch} \qquad (9a).$$

Here, $L_{max}$ is the travel distance to the outlet from the furthest point in the catchment, $\theta$ and $H$ are the
exponents in Flint's Law and Hack's Law respectively, and $k_C$ is a constant that lumps together $\theta$, H and
other factors, as shown in the appendix.
For the valley axis upstream of the channel head, from $x_{ch}$ to $L_{max}$, the elevation profile can be
written as follows (see appendix for derivation):
$$z_C = k_C \left[ \left( L_{\max} \right)^{1-\theta H} - \left( L_{ch} \right)^{1-\theta H} \right] + S_h \left( x - x_{ch} \right) \qquad \text{for } x_{ch} < x \leq L_{\max} \qquad (9b)$$



Here, $L_{ch}$ is the distance from the channel head to the outlet and $S_h$ represents a uniform slope over the
distance between $L_{ch}$ and $L_{max}$.

The upper boundary of the joint distribution of elevation and travel distance is defined by the

collection of points at the highest elevations in each travel distance bin. Unlike the channel profile, which
defines the base of the joint distribution, the points at the upper boundary do not necessarily lie along a
contiguous path. Nevertheless, for simplicity we refer to these points as the ridge profile, and assume that
its elevation follows a simple power-law relationship with distance.
$$z_R = k_R x^P \tag{10}$$

Here $k_R$ is an adjustable parameter and the exponent $P$ depends on the parameters of the channel profile. As
elaborated in the appendix, we impose the constraints that the ridge profile intersects the main-stem
channel profile at the two end points, where $x = 0$ and $x = L_{max}$, in order to define the parameter $P$.
**4.3 Scaling between area and relief**

Equations 9 and 10 provide the values of $z_C$ and $z_R$ that are needed in Equation 7 to define the local

relief for any travel distance. However, before Equation 7 can be used to generate synthetic distributions of
elevation and travel distance, the area in each travel distance bin ($A_L$) must be defined. We do so using the
previously discussed self-similar relationship between area and local relief shown in Figure 7C, where the
fraction of the total area in a travel bin of interest is proportional to the local relief divided by the sum of
local relief over all travel distance bins. For Inyo Creek, this relationship holds for any choice of bin
spacing and it is expressed mathematically in Equation 11
$$\frac{A_L}{A_C} = \frac{A_L}{\displaystyle\sum_{L=1}^{N} A_L} = \frac{R_L}{\displaystyle\sum_{L=1}^{N} R_L} \tag{11}.$$

Here, $N$ is the number of bins, $A_C$ is the catchment area, which is equal to the sum of all $A_L$, and $R_L$ is the
relief in the travel distance bin, which is equal to $z_R$-$z_C$. Following Hack's Law, the total area of the
catchment ($A_C$) can be treated as a power function of $L_{max}$ (see appendix).
**4.4 Generating synthetic distributions of elevation and travel distance**

Equations 7, 9, 10 and 11 can be used to generate fully synthetic distributions of elevation and

travel distance that are coupled to fundamental scaling relationships of natural catchments (expressed in
Hack's and Flint's laws). Moreover, this permits us to tune parameter values to reproduce catchments of
specific sizes and shapes. For example, Figure 10 shows the synthetic joint distribution of elevation and
travel distance for a catchment with size and shape similar to Inyo Creek (see appendix for the list of model
parameters used to generate this plot). By projecting the joint distribution of elevation and travel distance





onto the two orthogonal axes, we obtain the hypsometric curve and width function for the synthetic
catchment (Fig. 10, panels B and C). Thus, although the hypsometry and width function cannot be used
alone or together to generate the joint distribution of elevation and travel distance, they can be derived from
it. Nash-Sutcliffe statistics calculated from a comparison of the fully synthetic (Fig. 10A) and true
distribution (Fig. 4D) vary with bin size as in the previous case using the actual channel and ridge profiles,
as shown in Figure 9.  However, NS values for a given binning scheme are generally lower. This result
suggests that the fully synthetic formulation is less efficient than the partly synthetic formulation of section
4.1 at explaining variance in the joint distribution of elevation and travel distance. This loss of efficiency
arises due to error in fitting the upper and lower boundaries with the channel and ridge profile curves of
Equations 9 and 10.
**5. Discussion**
**5.1  Extending the model to other catchments**
The fully synthetic formulation for the joint distribution of elevation and travel distance was
calibrated using data from Inyo Creek, under the assumption that it is a prototypical catchment. But Inyo
Creek is relatively small and steep. This raises the question of whether the synthetic formulation yields
realistic results in other landscapes with lower relief or higher area.
Our other two study catchments, Providence Creek and Noyo River have lower relief and greater
area, respectively (Fig. 1). Hence we can use them to gauge the performance of the synthetic formulation
across a range of conditions. First we evaluated how well the beta distribution can be used as a predictor of
the distribution of elevation at each travel distance. Results are shown in Figure 11, which displays
normalized area-versus-elevation distributions for Providence Creek and Noyo River together with the
best-fit beta distributions for each catchment (with travel distance and elevation binned at 1/20 of
maximum values). The central tendency, spread, and skew of the best-fit beta distributions all appear to
roughly follow the patterns exhibited in the data.  However, the values of the best-fit shape parameters
differ between these two catchments, as well as with Inyo Creek for this binning scheme.  This suggest that
the joint distribution of travel distance and elevation, as represented by these model parameters, may differ
systematically between catchments.
The three catchments we analyzed vary across gradients in relief and drainage area (Fig. 1), but
also in the degree of dissection and channel profile shape, which may in turn reflect differing lithologic,
tectonic or climatic boundary conditions. For example, Providence Creek has a pronounced step in the
channel profile, with greater local relief and area concentrated in the upper part of the catchment (Fig. 2).
This step may arise due to feedbacks between weathering of biotite and topographic slope across the
landscape (Wahrhaftig, 1965). As a result, the channel profile is not well-fit by a power equation or any
other simple function.  In contrast, the larger Noyo River catchment has a smooth, highly concave main-



stem channel profile, and greater area at longer travel distances to the outlet due to a high degree of channel
branching.  The Noyo River main-stem channel profile may be influenced by aggradation due to sea-level
rise, and is better represented in the fully synthetic model using an exponential equation instead of a power
equation (see appendix).

Another second way to gauge model performance for various catchments is to compare predicted

hypsometric curves and width functions using the projections of the modeled and measured joint
distributions onto the elevation and travel distance axes, as we did in Fig. 10 for the fully synthetic Inyo
Creek case.  Figure 11 shows hypsometric curves and width functions for the three study catchments
generated with the DEM data ('actual'), the partially-synthetic formulation using actual profiles and
modeled area distributions (Eqns. 7 and 11), and the fully-synthetic formulation using modeled profiles.
For Inyo Creek, both the partly and fully synthetic models provide good fits to the overall shape of the
actual hypsometry and width function (Fig. 11a-b). In contrast, at Providence Creek, the partly synthetic
model only captures the hypsometry and width function over portions of the distributions, and performs
particularly poorly in the wide upper part of the catchment (Fig. 11c-d). Meanwhile, the fully synthetic
model performs more poorly because the modeled channel profile fails to capture the step in the
topography (Fig. 11 c-d). At Noyo River, despite its larger area, both the partly and fully synthetic models
perform reasonably well over all elevations and travel distances. Together these results suggest that both
the hypsometry and the width function of a wide range of catchments can be approximated to first order
using the framework developed here, provided that variations in the channel profile can be modeled.
**5.2 Future research opportunities**

Our results suggest many potentially fruitful avenues for future research.  First, joint distributions

of travel distance and elevation, combined with knowledge of rates of precipitation, erosion or other
material fluxes, can be used to understand how energy is created and dissipated across landscapes.  The
concept of source-area power provides a quantitative measure of the spatial distribution of processes that
influence the supply of materials to the catchment outlet. For example, this framework can be used to
understand how the size distribution of sediments passing through the catchment outlet is influenced by
weathering conditions at source elevations (Riebe et al., 2015), and by particle breakdown in transport
(Attal and Lave, 2009).  Catchment power, the integral of source-area power over the whole catchment,
provides a metric for comparisons between catchments, and could be used to quantify, and help explain, the
variation in topography across gradients in climate, tectonics and lithology.

A second set of research questions emerges from our approach to modeling synthetic joint

distributions of elevation and transport distance.  What explains the common tendency for positive skew in
the distribution of area with elevation for a given travel distance?  What do differences in the strength of
this asymmetry from one catchment to another tell us about landscape-forming processes? Why are area
and local relief within a travel distance bin linearly proportional, and does this relationship hold across a





wider suite of catchments? Can the model of a fully synthetic catchment be used to represent landscapes
across greater ranges of relief and drainage area than explored here?
Finally, the apparent success of our empirical model in capturing the bulk trends in the joint
distribution of elevation and travel distance in our study catchments suggests that there may be value in
developing a more comprehensive model, which accounts explicitly for the branching structure of the
channel network.  Such a model might have at its core a representation of the distribution of elevation and
travel distance for a first-order catchment similar to our empirical model for Inyo Creek. The model would
then represent larger catchments as combinations of multiple first-order headwater sub-catchments, and the
hillslope facets that drain directly to higher-order channel segments.  This raises the question of whether
there is a characteristic distribution of elevation for a given travel distance in the facets draining higher-
order valley slopes, and does it differ from the headwater sub-catchments in the same landscape?  Variation
in the topology of branching networks will shift the relative contributions of headwater sub-catchments and
higher-order facets to the number of source-areas at a given elevation or travel distance. How sensitive are
the distributions of source-area power to variations in network topology? Ultimately, such a model may
help explain both the central tendency and variability in the joint distribution of elevation and travel
distance, and provide a stronger theoretical foundation for understanding both the three-dimensional
structure of catchment topography.
**6 Summary**
Here we showed that the joint distribution of elevation and travel distance provides fresh
perspective on the vertical and horizontal structure of catchments in mountain landscapes (Fig. 4). In
particular, we showed that the paired values of elevation and travel distance can be collapsed into a single
index – the mean slope along the travel path – which varies both within and across catchments (Fig. 5).
Mean slope can be combined with knowledge of the fluxes and density of materials produced at, or
delivered to source areas, to define source-area power, and its integral catchment power, new metrics for
quantifying spatial variations in hydrologic and geomorphic processes within and between catchments (Fig.
6). To enable modeling of processes influenced by source-area power, we developed an empirical statistical
framework for defining the joint distribution of elevation and travel distance.  We used the Inyo Creek
catchment as a prototype, and found that the distribution of elevation between the main-stem channel and
ridge profiles, for a given travel distance bin, is well-represented by a parameterization of the beta
distribution.  To define a fully synthetic catchment, we derived power-law and exponential expressions for
the channel and ridge profiles, which when combined with the model for elevation distribution, can
produce realistic hypsometric curves and width functions. Key questions emerging from this work include:
how do patterns of source-area and catchment power vary across spatial gradients in climate, tectonics and
lithology?  What explains the characteristic skew of elevation distributions for a given travel distance? And



how do the distributions of source-area and catchment power arise from the branching properties of
networks and the relief structure of landscapes.
**Appendix A: Derivation of channel and ridge profile equations**
**A.1 Main-stem channel power-law profile**

To create an expression for the longitudinal profile of the main-stem channel, we coupled the

widely observed power-law scaling between slope ($S$) and drainage area ($A$)
$$S = k_s A^{-\theta} \tag{A1}$$
and the likewise common power-law scaling of main-stem distance ($L$) and area
$$A = k_A L^{H} \tag{A2}$$
In Equation A1, known as Flint's law, $k_s$ and $\theta$ are empirical coefficients (where $\theta$ is referred to as profile
concavity). In Equation A2, a version of Hack's law, $L$ is a local distance downstream from the catchment
divide along the main-stem valley axis, and $k_A$ and $H$ are empirical coefficients (with $H$ the reciprocal of the
Hack exponent). Hack's law can also be written in terms of the local travel distance upstream of the
catchment outlet, $x$,
$$A = k_A \left( L_{max} - x \right)^{H} \tag{A3}$$
where $L_{max}$ is the value of $L$ at the outlet (i.e., $x = L_{max} - L$).

Combining equations A1 and A3 we obtain an expression for mainstem channel slope, $S_C$, as a

function of distance upstream $x$
$$S_c = \frac{\partial z_c}{\partial x} = k_s k_A^{-\theta} \left( L_{max} - x \right)^{-\theta H} \tag{A4}$$
where $z_c$ is the elevation of the mainstem channel.

Integrating equation A4 provides an expression for the mainstem longitudinal profile

$$z_C = k_C \left[ \left( L_{max} \right)^{1-\theta H} - \left( L_{max} - x \right)^{1-\theta H} \right] \tag{A5a}$$
where
$$k_C = \frac{k_s k_A^{-\theta}}{1 - \theta H} \tag{A5b}$$





With the constraints that the elevation of the ridge $z_r$ and the channel $z_c$ match where $x = 0$ and $x =$

$L_{max}$, we can solve for the exponent $P$ as follows:
$$P = \frac{\log\left(z_{c\_max}/k_R\right)}{\log\left(L_{\max}\right)} \tag{A13}$$

Thus, the ridge network and the channel network are pinned together at the two end points.
**A.3 Inyo Creek power-law profile parameters**

The combined model for the ridge and channel profiles has 6 parameters; all other values are

calculated from the equations above. For the Inyo Creek channel and ridge profiles extracted from the
distributions of elevation for travel distances binned in 50 meter increments, Table A1 lists one possible set
of values that adequately reproduce the observed profile. These values were tuned to satisfy the following
constraints: $L_{max} = 4700$ m, the range of travel distances of Inyo rounded to nearest 50 m; drainage area at
outlet = 3.4 km$^2$; maximum elevation above outlet of 1890 m
**A.4 Main-stem channel exponential profile**

Exponential profiles have been used by many, including Hack (cites). Simply state elevation of the

channel as increasing exponentially with distance upstream of the outlet
$$z_c = k_e e^{\lambda x} \tag{A14}$$

where $k_e$ and lambda are empirical coefficients. As with the power profile, this is only valid between the
outlet and the channel head, where for simplicity we assume the slope becomes uniform. For the
exponential profile (equation A14), the channel slope
$$S_c = \frac{\partial z}{\partial x} = \lambda k_e e^{\lambda x} \tag{A15}$$

grows too slowly with increasing distance upstream of the channel head to represent the steep headwater
valley axis slope, so we define Sh-exp as an independent empirical model constant, with the constraint is
that it must be greater than the slope of the exponential profile at the channel head
$$S_{h\_exp} > S_{c\_max} = \lambda k_e e^{\lambda(L_{\max} - L_{ch})} \tag{A16.}$$

The full channel profile expression becomes
$$z_c = k_e e^{\lambda x} \qquad\qquad \text{for } 0 \le x \le x_{ch} \tag{A17a}$$





$$z_c = k_e e^{\lambda x_{ch}} + S_{h\_exp}\left(x - x_{ch}\right) \qquad \text{for } x_{ch} < x \le L_{max} \tag{A17b}$$

and the highest point along the mainstem profile, $Z_{C\_max}$ is
$$z_{c\_max} = k_e e^{\lambda x_{ch}} + S_{h\_exp} L_{ch} \tag{A18}.$$

**A.5 Ridge exponential profile**
To define the ridge long profile, for symmetry with the channel profile we assume an exponential
relation between elevation and distance,
$$z_R = k_{Re} e^{\gamma x} \tag{A19}$$

Where the coefficient $k_{Re}$ is an adjustable parameter, and the exponent $\gamma$ depends on the parameters of the
channel profile. As with the power law profile derivation, to specify $\gamma$ we impose the constraints that the
ridge profile must intersect the mainstem channel profile at the two end points, where $x = 0$ and $x = L_{max}$,
the lowest and highest points in the landscape.
With the constraints that the elevation of the ridge $z_r$ and the channel $z_c$ match where $x = L_{max}$, we
can solve for the exponent $\gamma$
$$\gamma = \frac{\ln\left(z_{c\_max}/k_{Re}\right)}{L_{max}} \tag{A20}$$

The ridge network and the channel network are pinned together at these two end points.
**A.6 Inyo Creek exponential profile parameters**
The combined model for the two exponential profiles has five parameters; all other values are
calculated from the equations above. Table A2 lists one possible best fit (by eye) set of values for the Noyo
River channel and ridge profiles extracted from the distributions of elevation for travel distances binned in
250 meter increments. These values were tuned to satisfy the following constraints: $L_{max} = 20,750$ m, the
range of travel distances of Inyo rounded to nearest 50 m; maximum elevation above outlet = 620 m (along
mainstem profile).
**Data Availability**
The DEMs used in this paper can be obtained upon request from the corresponding author.



**Acknowledgments**
We thank Sarah Konrad and Catherine Noll for contributing to preliminary DEM analysis. Funding was
provided by the Doris and David Dawdy Fund for Hydrologic Research, and National Science Foundation
Grants EAR 1325033 and 1239521.

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

**Figure captions**
**Figure 1. Study site locations and comparison of channel and ridge profiles.** Left: Location map of
study catchments in California, USA. Right: Profiles of the lowest point at each travel distance (i.e., the
mainstem channel) and the highest point at each travel distance (referred to here as the ridge profile). The
channel and ridge profiles enclose all paired values of elevation and travel distance for each catchment.
Differences in catchment relief and size across the sites produce distinct populations of paired values. The
ratio of elevation to travel distance is the mean slope along a path from the source to the catchment outlet.
Thus the catchments also harbor distinct populations of mean slope.
**Figure 2. Spatial distributions of elevation and travel distance.** Maps showing the spatial distribution of
elevation and travel distributions across the Inyo Creek (A), Providence Creek (B), and Noyo River (C)
study catchments. Black lines are elevation contours, with hillshade in background for emphasis. Color
shade shows scaled values of travel distance (normalized by the maximum value in the catchment). Note
variation in scale and compass orientation from one watershed to the next. Elevation contour spacing is 50
m in (C) and (B), and 200 m in (C).
**Figure 3. Hypsometry and width functions.** Normalized frequency distributions of elevation (a) and
travel distance to the outlet (b). Frequencies are normalized so that the area under the curve is equal to 1 in
each case. Binning increment is 1/47 of maximum value (Table 1).
**Figure 4. Joint distributions of elevation and travel distance.** Distribution of source area elevations and
travel distances from 10 m DEMs of catchments drained by (a) Inyo Creek, (b) Providence Creek, and (c)
the Noyo River. Bivariate frequency distributions of elevation and travel distance for each catchment (d-f)
show relative density (color bar in (d); data binning as in Figure 2.
**Figure 5. Distribution of mean slope across catchments.** Histograms (insets, A-C) of mean slope along
travel path from source to outlet (ratio of source area elevation to travel distance), with colors highlighting
bins of relatively low, medium and high values. Bins of common mean slope form linear bands on plots of
elevation versus travel distance (A-C). Maps of catchments (D-F) show spatial distribution of source areas
sharing similar mean slope for highlighted values.



**Figure 6. Spatial distribution of source-area power for water and sediment.** Histograms (left) of
source-area power calculated using equation 3 for the Inyo Creek catchment for water delivered by
precipitation (A), and sediment produced by erosion (B). Panel (C) shows dimensionless ratio of source-
area water power to sediment production rate (eqn. 4); colors highlight bins of relatively low, medium and
high values. Maps (right) show spatial distribution of highlighted values. Note the sharp increase in water
power per sediment flux from upper to lower parts of the catchment.
**Figure 7. Elevation distributions for different travel distances at Inyo Creek.** (A) Elevation data points
for Inyo Creek catchment parsed into forty-seven 100-m wide travel distance bins. (B) Distributions of
elevation for seven representative travel distance bins; colors correspond to shaded bins in panel A, mean
travel distance indicated for each curve. (C) Fraction of total area in each travel distance bin as a function
of fraction of total relief in each bin, roughly follows 1:1 line, colored symbols indicate representative bins
in panels A and B. (D) Collapse of elevation distributions for each travel distance bin, with elevation
normalized by relief within bin and area by total area within bin. Best-fit beta distribution captures typical
shape of hypsometry for a given travel distance.
**Figure 8. Objective function for best-fit beta distribution shape parameters.** Contour plot of root mean
sum of squared error (RMSE) between actual and predicted area density of elevation for a given travel
distance for paired values of beta distribution shape parameters. Minimum RMSE at $\alpha = 2.6$ and
$\beta = 3.4$ as indicated by diamond. In this example, travel distance and elevation bin sizes equal 100 m and
40 m respectively.
**Figure 9. Model performance**. Variation in Nash-Sutcliff model efficiency statistic (Eqn. 8) with size of
travel distance and elevation bins, for modeled joint distributions of elevation and travel distance for Inyo
Creek, using actual profiles (solid lines) and modeled profiles (dashed lines). Nash-Sutcliff value of 1.0
indicates perfect agreement between modeled and actual distribution of area; value of 0 indicates model
performance no better than uniform distribution of mean area density. A trade-off between model
efficiency and spatial resolution is revealed by trend toward higher Nash-Sutcliff values for larger bin sizes.
**Figure 10. Normalized Distribution of elevation by travel distance bin for other catchments**. Travel
distance and elevation bin sizes = 1/20 of maximum values  Thin lines show elevation distributions,
normalized by local relief, for each travel distance bin. Thick colored curves show best-fit beta
distributions, with shape parameter values indicated. Normalized elevation distributions are more skewed
for Noyo River, reflecting larger drainage area and greater degree of landscape dissection.
**Figure 11. Fully synthetic joint distribution of elevation and travel distance for catchment the size of**
**Inyo Creek**. In (A) channel and ridge profiles are defined by equations 9 and 10, area density (color bar)
given by equations 7 and 11. Side panels show area density projected on distance axis to create width
function (B) and projected on elevation axis to create hypsometric curve (C).



**Figure 12. Comparison of actual with modeled hypsometric curves and width functions for three**
**study catchments.** In each panel, thick colored curves show data from catchment DEM, while thick and
dashed black lines show model predictions using actual and modeled channel and ridge profiles
respectively. Also shown in left panels are hypsometric curves predicted using uniform area distribution,
for the case when Nash-Sutcliff model efficiency statistic = 0; for this case, predicted width function
matches actual.




**Table 1. Study site characteristics**

|  | Inyo Creek | Providence Creek | Noyo River |
|---|---|---|---|
| Drainage Area (km$^2$) | 3.4 | 8.1 | 144 |
| Relief (m) | 1,895 | 1,117 | 893 |
| Max Travel Distance (m) | 4,660 | 7,940 | 20,790 |
| Mean Slope to outlet | 0.33 | 0.14 | 0.021 |
| Elevation of outlet (masl) | 2053 | 998 | 84 |
| Outlet UTM North | 392369.717 | 300456.028 | 364182.531 |
| Outlet UTM East | 4049943.32 | 4101509.08 | 450994.25 |

**Table A1. Inyo Creek power-law profile model parameters**

| Parameter | Value |
|---|---|
| $\theta$ | 0.31 |
| $H$ | 1.75 |
| $k_s$ | 25 |
| $k_A$ | 1.28 |
| $L_{ch}$ | 600 m |
| $K_R$ | 0.6 |

**Table A2. Noyo River exponential profile model parameters**

| Parameter | Value |
|---|---|
| $\lambda$ | $1.8 \times 10^{-4}$ m$^{-1}$ |
| $S_{h\_exp}$ | 0.16 |
| $k_e$ | 6.7 m |
| $L_{ch}$ | 2000 m |
| $K_{Re}$ | 195 m |

845





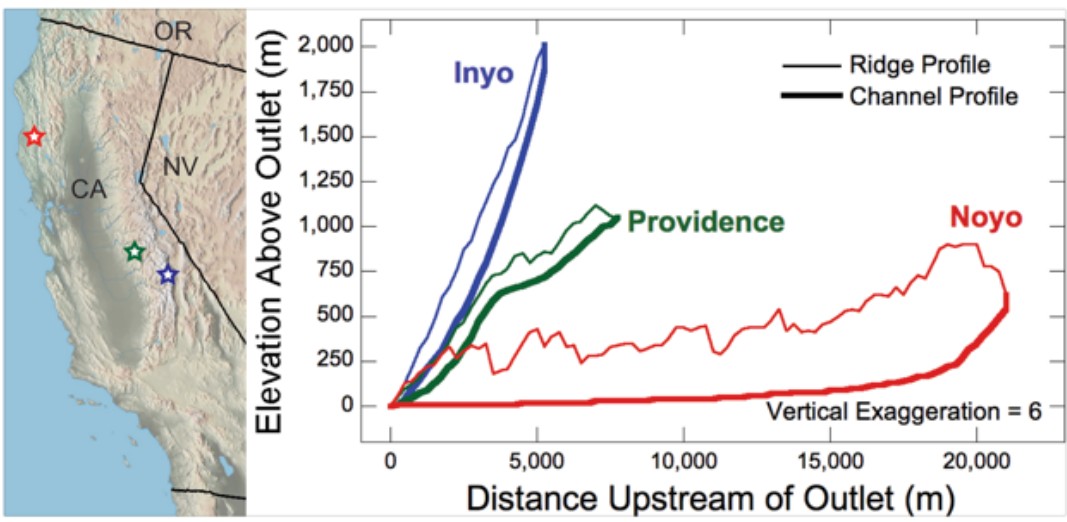

**Figure 1. Study site locations and comparison of channel and ridge profiles.**
Left: Location map of study catchments in California, USA. Right: Profiles of the
lowest point at each travel distance (i.e., the mainstem channel) and the highest point
at each travel distance (referred to here as the ridge profile). The channel and ridge
profiles enclose all paired values of elevation and travel distance for each catchment.
Differences in catchment relief and size across the sites produce distinct populations
of paired values. The ratio of elevation to travel distance is the mean slope along a
path from the source to the catchment outlet. Thus the catchments also harbor
distinct populations of mean slope.




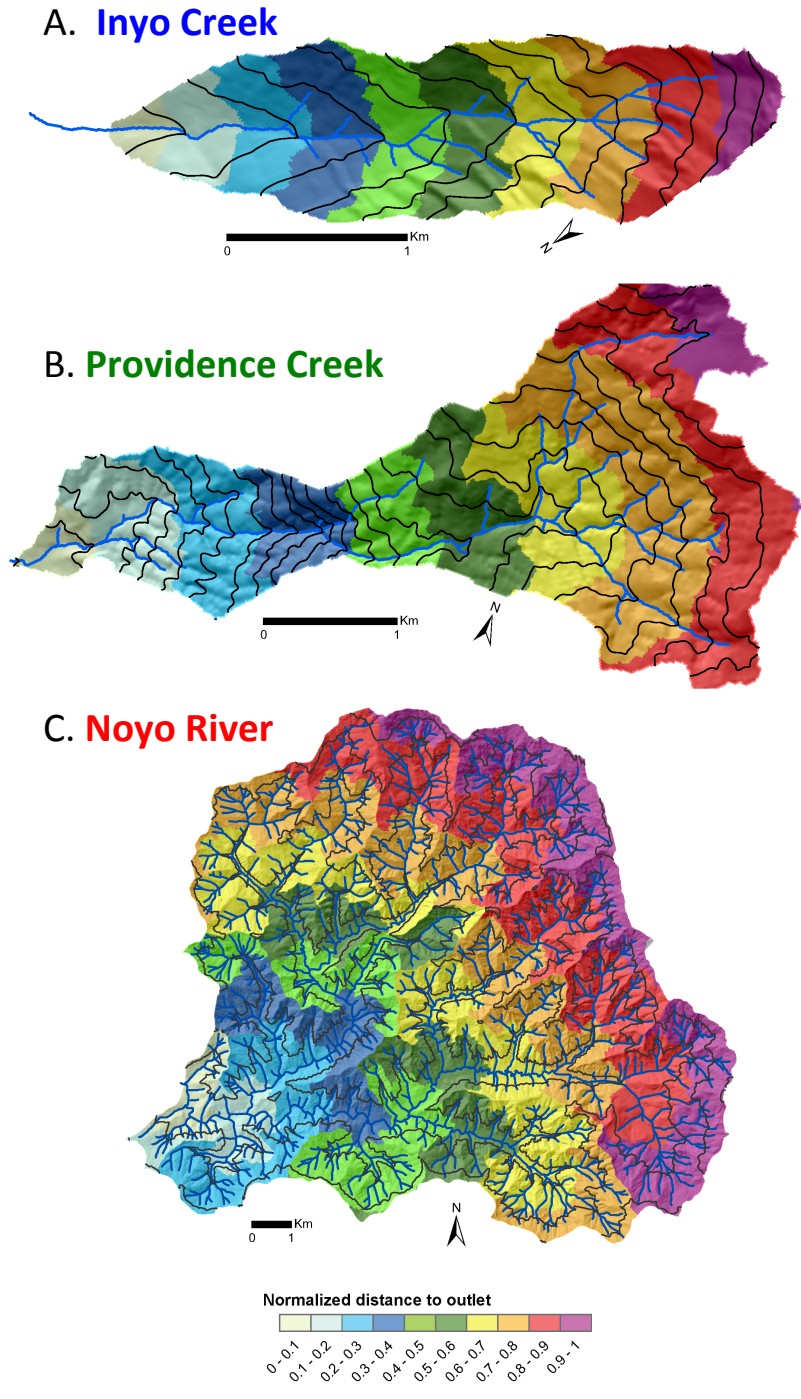

**Figure 2. Spatial distributions of elevation and travel distance.** Maps showing the spatial distribution of elevation and travel distributions across the Inyo Creek (A), Providence Creek (B), and Noyo River (C) study catchments. Black lines are elevation contours, with hillshade in background for emphasis. Color shade shows scaled values of travel distance (normalized by the maximum value in the catchment). Note variation in scale and compass orientation from one watershed to the next. Elevation contour spacing is 50 m in (C) and (B), and 200 m in (C).



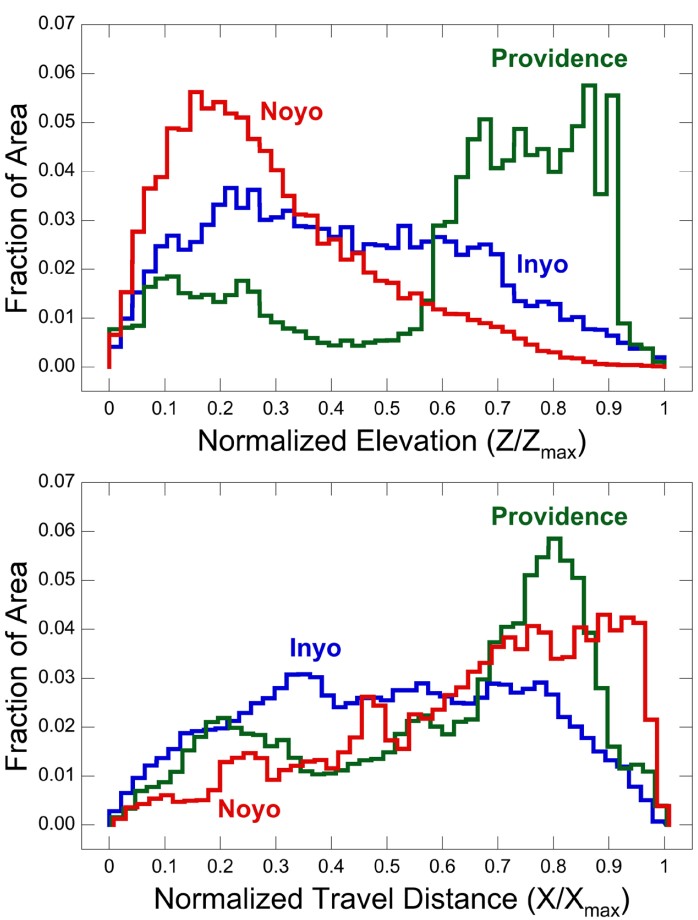

**Figure 3. Hypsometry and width functions.** Normalized frequency distributions of elevation (a) and travel distance to the outlet (b). Frequencies are normalized so that the area under the curve is equal to 1 in each case. Binning increment is 1/47 of maximum value (Table 1).



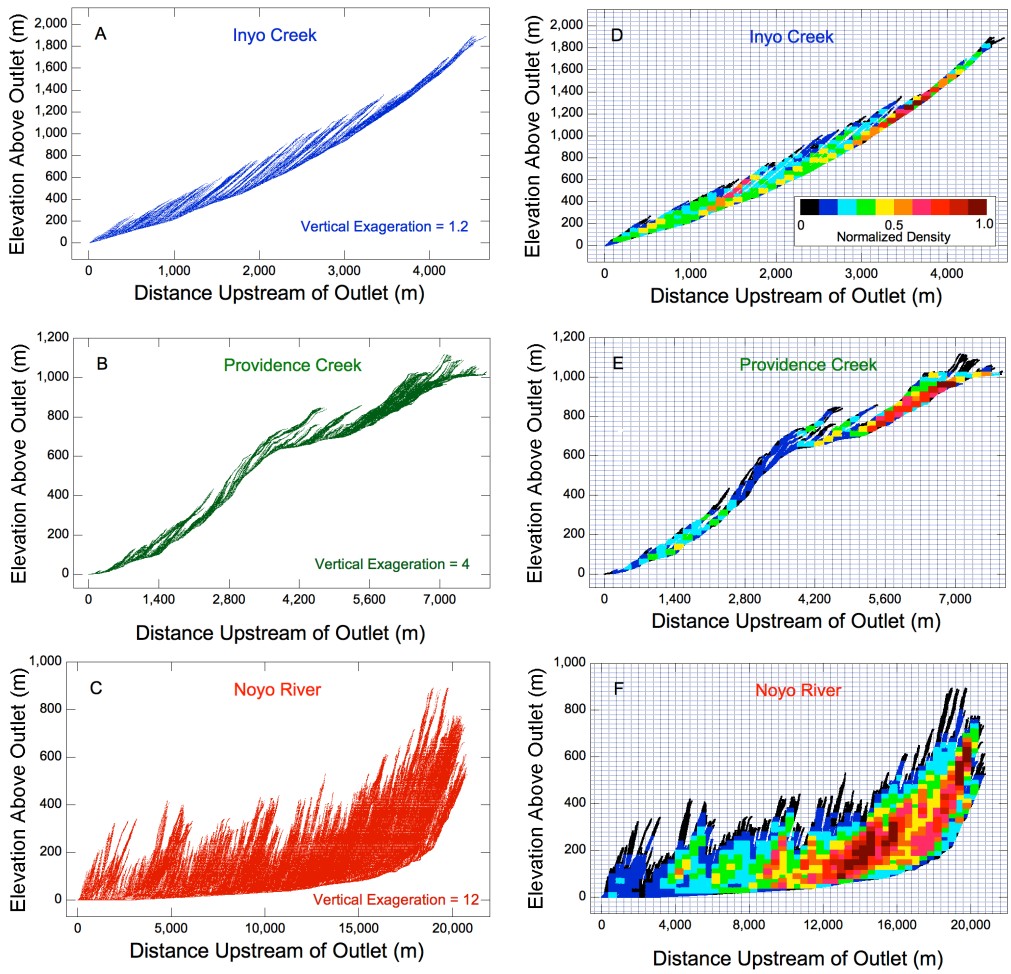

**Figure 4. Joint distributions of elevation and travel distance.**
Distribution of source area elevations and travel distances from 10 m
DEMs of catchments drained by (a) Inyo Creek, (b) Providence Creek,
and (c) the Noyo River. Bivariate frequency distributions of elevation
and travel distance for each catchment (d-f) show relative density (color
bar in (d); data binning as in Figure 2.




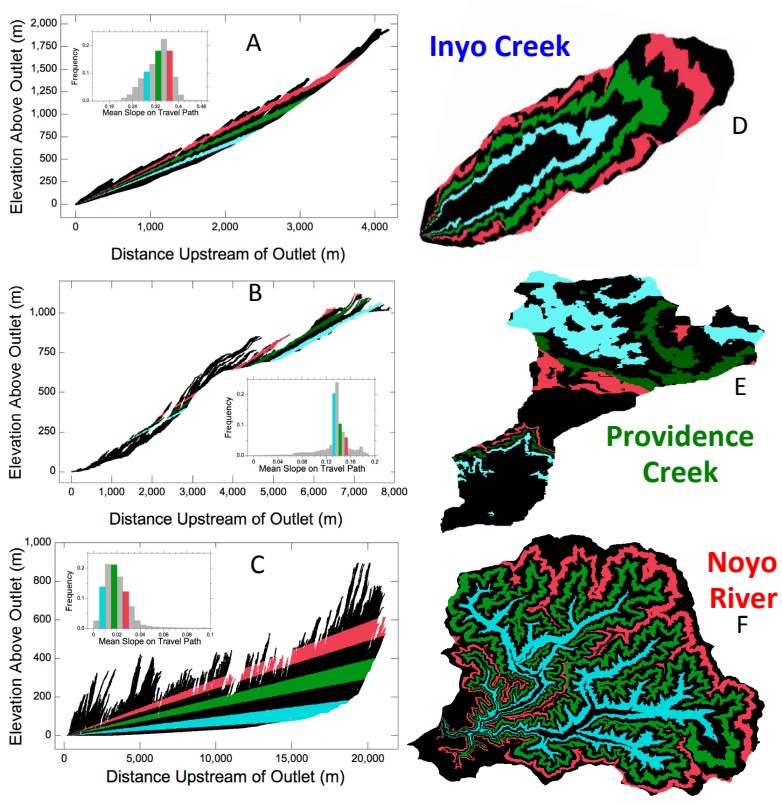

**Figure 5. Distribution of mean slope across catchments.** Histograms (insets, A-C) of mean slope along travel path from source to outlet (ratio of source area elevation to travel distance), with colors highlighting bins of relatively low, medium and high values. Bins of common mean slope form linear bands on plots of elevation versus travel distance (A-C). Maps of catchments (D-F) show spatial distribution of source areas sharing similar mean slope for highlighted values.




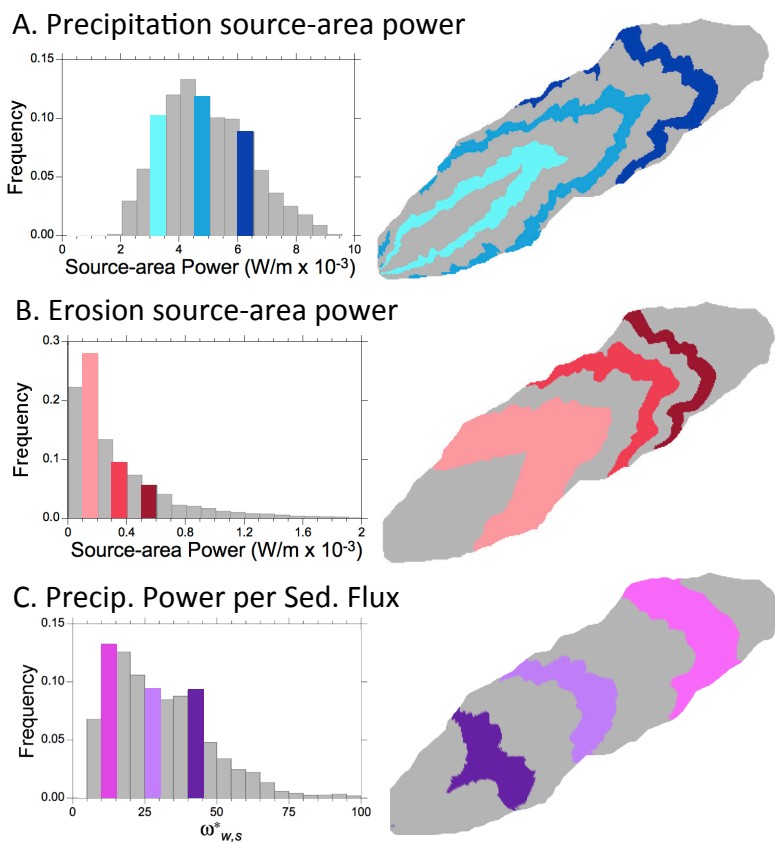

**Figure 6. Spatial distribution of source-area power for water and sediment.**
Histograms (left) of source-area power calculated using equation 3 for the Inyo
Creek catchment for water delivered by precipitation (A), and sediment produced
by erosion (B). Panel (C) shows dimensionless ratio of source-area water power to
sediment production rate (eqn. 4); colors highlight bins of relatively low, medium
and high values. Maps (right) show spatial distribution of highlighted values. Note
the sharp increase in water power per sediment flux from upper to lower parts of
the catchment.



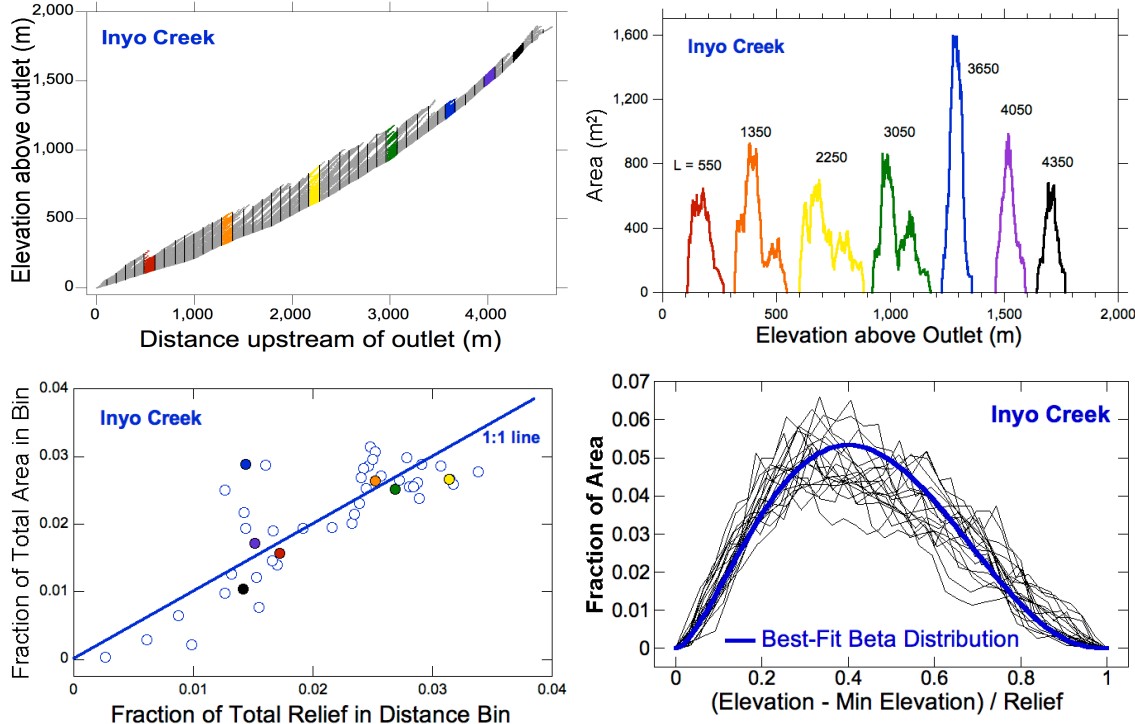

**Figure 7. Elevation distributions for different travel distances at Inyo Creek**
(A) Elevation data points for Inyo Creek catchment parsed into forty seven 100-m
wide travel distance bins. (B) Distributions of elevation for seven representative travel
distance bins; colors correspond to shaded bins in panel A, mean travel distance
indicated for each curve. (C) Fraction of total area in each travel distance bin as a
function of fraction of total relief in each bin, roughly follows 1:1 line, colored
symbols indicate representative bins in panels A and B. (D) Collapse of elevation
distributions for each travel distance bin, with elevation binned in 40 m increments.
Elevation is normalized by total relief within distance bin and area normalized by total
area within bin. Best-fit beta distribution captures typical shape of hypsometry for a
given travel distance.





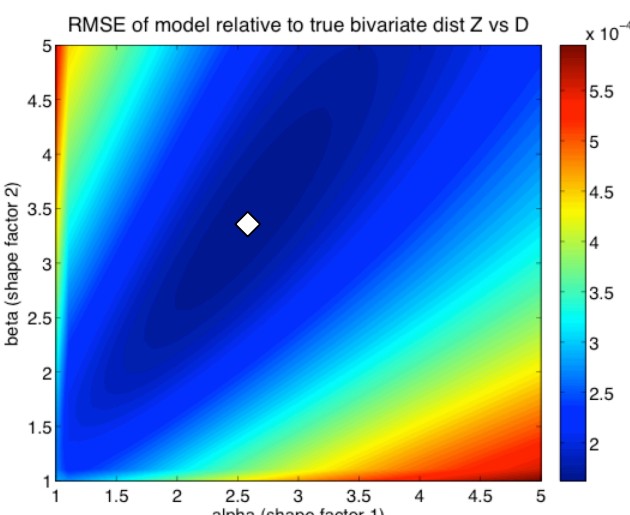

**Figure 8. Objective function for best-fit beta distribution shape parameters**
Contour plot of root mean sum of squared error (RMSE) between actual and predicted area
density of elevation for a given travel distance for paired values of beta distribution shape
parameters. Minimum RMSE at $\alpha = 2.6$ and $\beta = 3.4$ as indicated by diamond. In this example,
travel distance and elevation bin sizes equal 100 m and 40 m respectively.





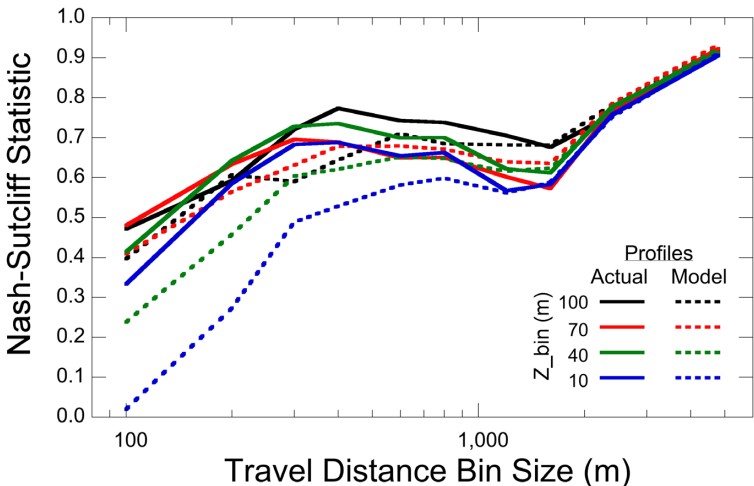

**Figure 9. Model performance**. Variation in Nash-Sutcliff model efficiency statistic with size of travel distance and elevation bins, for modeled joint distributions of elevation and travel distance for Inyo Creek, using actual profiles (solid lines) and modeled profiles (dashed lines). Nash-Sutcliff value of 1.0 indicates perfect agreement between modeled and actual distribution of area; value of 0 indicates model performance no better than uniform distribution of mean area density. A trade-off between model efficiency and spatial resolution is revealed by trend toward higher Nash-Sutcliff values for larger bin sizes.





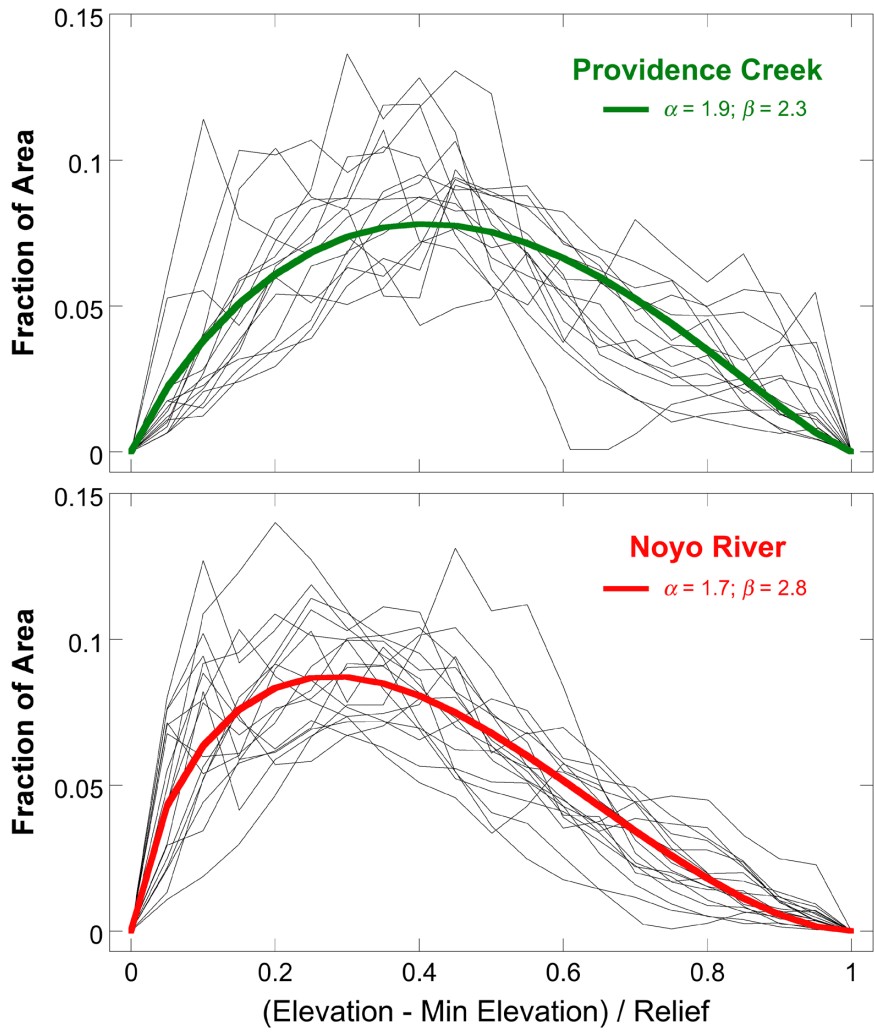

**Figure 10. Normalized Distribution of elevation by travel distance bin for other catchments.**
Travel distance and elevation bin sizes = 1/20 of maximum values  Thin lines show elevation
distributions, normalized by local relief, for each travel distance bin.  Thick colored curves show
best-fit beta distributions, with shape parameter values indicated.  Normalized elevation distributions
are more skewed for Noyo River, reflecting larger drainage area and greater degree of landscape
dissection.





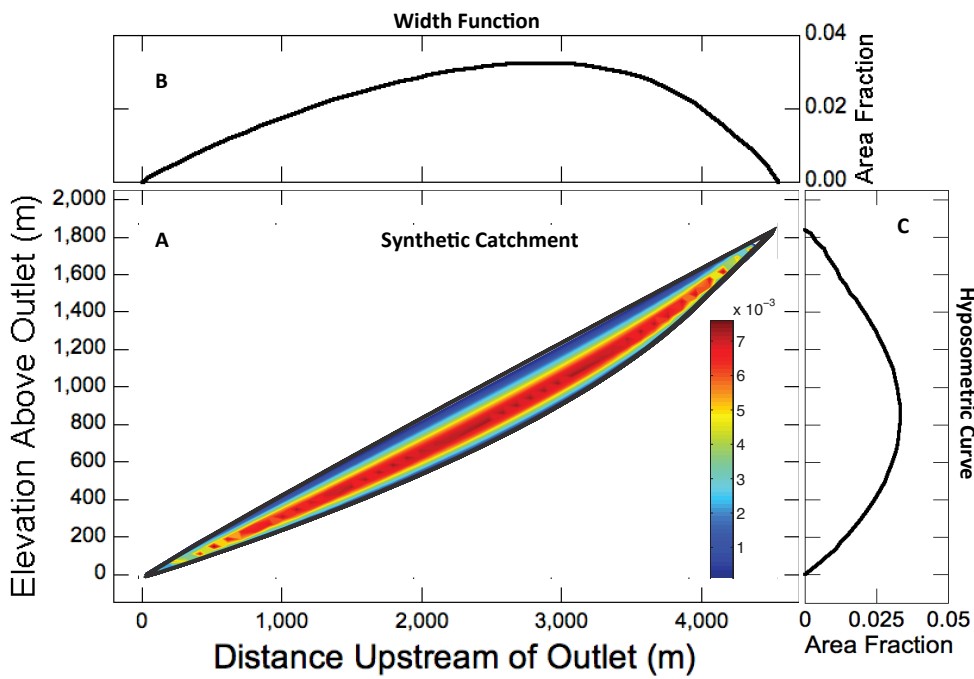

**Figure 11. Fully synthetic joint distribution of elevation and travel distance for catchment the size of Inyo Creek.** In (A) channel and ridge profiles are defined by equations XX and YY, area density (color bar) given by equation ZZ. Side panels show area density projected on distance axis to create width function (B) and projected on elevation axis to create hypsometric curve (C).





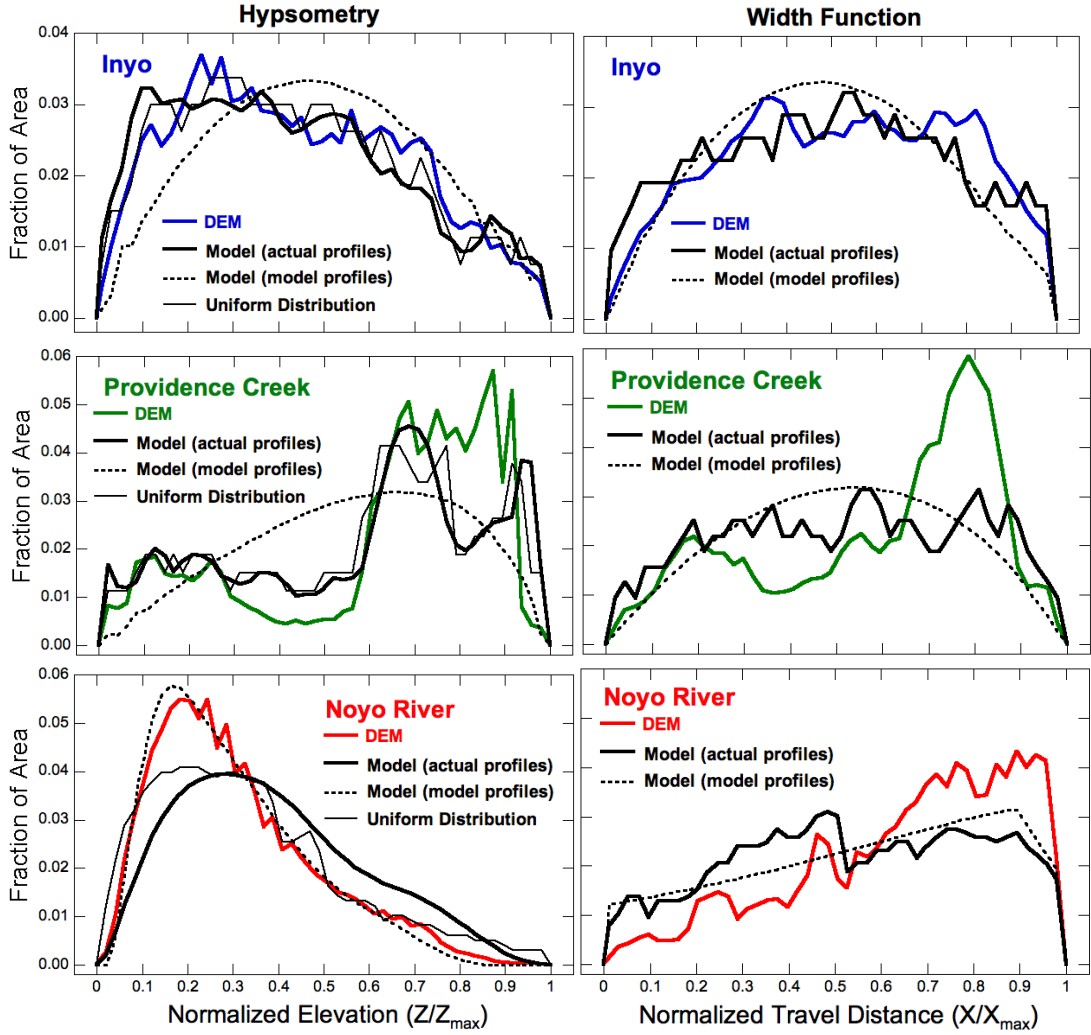

**Figure 12.  Comparison of actual with modeled hypsometric curves and width functions for three study catchments.**  In each panel, thick colored curves show data from catchment DEM, while thick and dashed black lines show model predictions using actual and modeled channel and ridge profiles respectively.  Also shown in left panels are hypsometric curves predicted using uniform area distribution, for the case when Nash-Sutcliff model efficiency statistic = 0; for this case, predicted width function matches actual.