# Peer review of "Catchment power and the joint distribution of elevation 1"

_Earth Surface Dynamics, 2016_

## Referee Comment (RC1) · Anonymous Referee #1 · 29 Mar 2016

This study attempts to understand the joint distribution of elevation and travel distance to the outlet and explore how it varies across a few study catchments which remarkably differ with respect to the width function and hypsometric curve. The joint distribution of elevation and travel distance is then used to define indices of "source-area power" and "catchment power", with the hope to express how varying rates of water and sediment transport throughout different catchments can be expressed by such metrics. Finally, an empirical algorithm is suggested for generating synthetic source-area power distributions to explore the effects of topography on the water and sediment fluxes passing through catchments.

Specific comments:

1) In some parts of the paper, it is claimed that the proposed methodology can be

used to answer some specific questions. Here are the examples: - Lines 23-23 saying that the empirical algorithm for generating synthetic source-area power distributions can be used to explore the effects of topography on the water and sediment fluxes passing through catchments. - Lines 64-65 saying that do the distributions of elevation and travel distance to the outlet differ in ways that systematically reflect the factors that drive landscape evolution, such as weathering, climate, and tectonics? - Lines 71-73 saying that if the synthetic catchments are able to explore how factors such as area, relief, and profile concavity influence catchment power. Unfortunately, none of the above questions are addressed in this paper, except a few qualitative explanations.

2) In figure 1, do the given profiles correspond to the longest flow path in those catchments? Also what extra information does this figure provide in comparison to figure 4?

3) Lines 118-120 are not clear at all from figure 2. Authors might want to clarify it directly in the figure 2.

4) In lines 166-167, it is said that "the joint distribution plots generally show dense concentrations of data points at low elevations for any given travel distance". This is not definitely true based on the color bar given in figure 4d. For instance in the Providence Creek, very small concentrations of data points exist at low elevations over a wide range of travel distances less than 4000 m. Similar observation can be made for the Inyo Creek for travel distances less than 3000 m, except some high concentration data points spanning around travel distances of 1500 m.

5) In lines 169-171, it is said that "for a given travel distance, as elevation decreases, data point density generally increases to a peak and then quickly tapers to zero." Should not it be as elevation increases? At least all the plots (4c, 4e, 4f) show that for any given travel distance, the data point density goes to zero at the highest elevations (depicted by black colors).

6) In lines 171-172, it is said that "they also show that the density of paired values is

highest at 60 and 80% of the maximum travel distance". This is not true at all except for the Noyo River, while the statement is given in a general sense.

7) First of all, direct comparison of figures 3 and 4 is not easy as the horizontal axes show the same quantity, but different ranges (authors might want to make it consistent throughout the paper). Second, figure 3 shows that at the Noyo River, the majority of the area pertains to long travel distances and low elevations. Can the authors explain why this is not reflected in figure 4f where highly dense data points correspond to mid travel distances and relatively high elevations?

8) In lines 231-233, it is said that "in landscapes where rates of precipitation and erosion are spatially variable and sometimes correlated, we expect the distributions of power and mean slopes to differ". Then the Inyo Creek catchment is mentioned as an example of this case. But comparing figures 5 and 6a does not support this at all, i.e., the spatial patterns of "water" power and mean slopes are very identical to each other in this catchment. How do the authors explain this?

9) In figure 12, it seems that the hypsometric curves and width functions generated with the partially-synthetic formulation using actual profiles fits better to the real data than the fully-synthetic formulation using modeled profiles in the Noyo River. But the reverse is observed for the Providence Creek. Can the authors explain why the partially-synthetic formulation using actual profiles should not always result in better fitting?

Technical corrections:

Line 438: figure 11 instead of 10

Line 444: figure 11A instead of 10A

Line 456: figure 12 instead of 11

Line 477: figure 11 instead of 10

Line 478: figure 12 instead of 11

**ESurfD**
Line 482: figure 12a-b instead of 11a-b

Line 484: figure 12c-d instead of 11c-d

Line 486: figure 12c-d instead of 11c-d

---

## Referee Comment (RC2) · Anonymous Referee #2 · 22 Jun 2016

**Review of manuscript by Sklar et al. for Earth Surf. Dynamics**

This paper proposes two new metrics to quantify landscape morphology based on the distribution of elevation and travel-distance, brought together in the concept of catchment power. Three examples of catchments with different morphologies are explored and a method is proposed by which artificial catchments with specified source-area power distributions can be synthesised.

The paper's methods are certainly novel and raise some important questions about the formation of landscapes and the topic will be of broad interest to readers of ESurf. The manuscript is well written and carefully presented. I suggest that the manuscript is suitable for publication in ESurf, subject to satisfactory additional clarification and discussion of the following points.

**General Comments**

- 1. The paper concludes by stating that its major contribution is to offer a "fresh perspective". That's fine, but it would be better in my view to explain what new knowledge is available through the use of the new landscape metrics. The reader is left unclear on how this particular set of metrics might shed light on important problems in geomorphology.
- 2. The calculation of stream power (line 224) takes as the relevant slope the mean slope along the path to the catchment outlet. If the actual slope is close to the mean slope then this may be a good approximation. If not (for example, if the pathway might involve a very steep upper section with a long flat floodplain, or alternatively a high elevation plateau with a steep ravine descending from it) then the virtual velocity of sediment through the system will differ substantially, with important implications for residence time of sediment in floodplains etc (which is itself relevant geochemical residence times in the catchment, cosmogenic methods, and carbon sequestration). This warrants some further discussion.
- 3. In section 3 (line 243 onwards) the notation switches from the generic subscripts i,j to w and s for water and sediment, and the dimensionless ratio \omega^\* is defined as the ratio of source-area power of water per mass of sediment. The intuitive/conceptual significant of this ratio is not clear, which makes it hard to interpret the values 36–653 in the subsequent paragraph.

**Specific comments / Minor points**

Line 86 Tarbotton -> Tarboton

---

## Author Response (AR1)

LEONARD S. SKLAR, PH.D. PROFESSOR OF GEOLOGY DEPARTMENT OF EARTH & CLIMATE SCIENCES

> 1600 Holloway Avenue San Francisco, CA 94132

Office: 415-338-1204 Fax: 415-338-7705 Email: leonard@sfsu.edu

Susan Conway, John Hillier, Giulia Sophia, Guest Editors Special issue on Frontiers in Geomorphometry Earth Surface Dynamics

August 5, 2016

Dear Colleagues,

This letter accompanies re-submission of the manuscript "*Catchment power and the joint distribution of elevation and travel distance to the outlet*," following the discussion phases, for final consideration for publication in the special issue on Frontiers in Geomorphometry.

Thank you for recruiting the two referees who provided helpful review comments during the discussion phase. We have carefully considered each of the comments and done our best to address every one. As detailed in the author responses attached in the pages that follow, we have made changes to the manuscript in response to each of the comments; following the detailed responses you will find a 'track changes' version of the revised manuscript that shows every change made during the revision process. Together these changes strengthen the paper and hopefully make the findings and interpretations as clear as possible to the wide range of readers who may be interested in this work.

My three coauthors and I are agreed that the revised manuscript is ready for resubmission. Thank you very much for your kind consideration.

Sincerely,

Leonard Sklar

Leonard Sklar, PhD

Author responses to referee comments on "Catchment power and the joint distribution of elevation and travel distance to the outlet" by L. Sklar et al.

**Author Response to Comments by Referee #1**

We would first like the thank Reviewer 1 for their constructive comments. In this response we provide answers to all the comments and detail the changes that will be applied in the revised manuscript. Please note that line numbers refer to the numbering of the original discussion manuscript.

**Comment 1:**

"In some parts of the paper, it is claimed that the proposed methodology can be used to answer some specific questions. Here are the examples: - Lines 23-23 saying that the empirical algorithm for generating synthetic source-area power distributions can be used to explore the effects of topography on the water and sediment fluxes passing through catchments. - Lines 64-65 saying that do the distributions of elevation and travel distance to the outlet differ in ways that systematically reflect the factors that drive landscape evolution, such as weathering, climate, and tectonics? - Lines 71-73 saying that if the synthetic catchments are able to explore how factors such as area, relief, and profile concavity influence catchment power. Unfortunately, none of the above questions are addressed in this paper, except a few qualitative explanations."

**Answer:**

We appreciate the reviewer's interest in seeing these proposed applications of the methodology. The statements referred to in the comment all occur in the abstract and the introduction, and contribute to the motivation for developing the model. It is beyond the scope of this paper to both develop and apply the model. Subsequent applications of the model are best done in subsequent papers. Since discussion of this paper began, already we have submitted and had accepted the first of several planned papers that apply the model to addressing the types of questions referred to in this comment. The paper is Lukens et al., "Grain size bias in cosmogenic nuclide studies of stream sediment in steep terrain", currently in press at the Journal of Geophysical Research – Earth Surface. In that paper, we use the catchment power framework, and the algorithm for generating synthetic distributions of elevation and travel distance developed here, for evaluating risk of bias is estimating catchment average erosion rates when sampling a single sediment size class (e.g. medium sand) given a spatial gradient in the size distribution of sediments produced on hillslopes. It would not have been practical to include the results of that application of the model within this paper. We will make changes to this manuscript to clarify for the reader that the applications of the model are expected to come in subsequent papers.

**Changes in the manuscript:**

Abstract passage beginning on Line 23 has been change to read "We then develop an empirical algorithm for generating synthetic source-area power distributions, which

can be parameterized with data from natural catchments. This new way of quantifying the three-dimensional geometry of catchments can be used to explore the effects of topography on the distribution on fluxes of water, sediment, isotopes and other landscape products passing through catchment outlets, and may provide a fresh perspective on problems of both practical and theoretical interest."

Introduction passage beginning on Line 71 has been changed to read "Next, using our analyses of the elevation and travel distance distributions from the study catchments, we develop an approach for generating synthetic catchments that capture many features of power distributions in natural landscapes. Finally, we discuss how our approach can be used to explore how factors such as area, relief, and profile concavity influence catchment power and more broadly how rivers are influenced by hillslope sources of water, solutes, and sediment (e.g. Lukens et al., 2016)."

**Comment 2:**

"In figure 1, do the given profiles correspond to the longest flow path in those catchments? Also what extra information does this figure provide in comparison to figure 4?"

**Answer:**

As explained in the figure 1 caption, what we refer to as the "mainstem profile" is the lowest elevation for that travel distance while the "ridge profile" is the highest elevation for that travel distance. The longest flow path in each catchment corresponds to the highest and most distant point shown. This figure has three purposes that set it apart from figure 4. It introduces the study catchments and their geographic locations, and provides a simple comparison of the relative scale of each in terms of both elevation and travel distance. It also graphically poses the key questions that we seek to answer in this paper, how do we fill in the blank space between these two types of profiles? In other words, what is the joint distribution of elevation and travel distance to the outlet, and how do these distributions differ between catchments? Figure 4 has a very different purpose. It begins to answer those questions by filling in the blank space between the profiles with data and calculated density distributions. In figure 4 it is not practical to plot the three catchments at the same scale, so the relative scales of the catchments must be shown elsewhere; we chose to do that right away in figure 1.

**Changes in the manuscript:**

The first part of the figure 1 caption has been changed to read: "Left: Location map of study catchments in California, USA. Right: Elevation profiles of the lowest point at each travel distance (i.e., the mainstem channel) and the highest point at each travel distance (referred to here as the ridge profile). The longest and shortest travel distances in each catchment are the points where the two profiles meet."

**Comment 3:**

"Lines 118-120 are not clear at all from figure 2. Authors might want to clarify it directly in the figure 2."

**Answer:**

The text in question reads "Conversely, for a given travel distance, elevations are highest at the ridges and lowest in the valley axis. These patterns are especially clear at Inyo Creek (Fig. 2a) and Providence Creek (Fig. 2b), which drain small, relatively undissected catchments." The best way to see this is to follow the boundary between two color bands, which represents a contour of fixed travel distance, and consider how the distance to the nearest elevation contour (black) line changes. Fig. 2a (Inyo Creek) is provides the clearest example. Consider the travel distance contour of 0.6 times the maximum travel distance, which is the boundary between the dark green and yellow color bands. An elevation contour line crosses this travel distance contour right at the valley axis. As one follows either the color band boundary or the elevation contour, the contours diverge. The elevation contour crosses higher elevation contour lines. With that example in mind, the same pattern can be discerned in the other two catchments, although the increase in scale introduces substantial variability.

**Changes in the manuscript:**

Starting with the sentence beginning at line 117, the passage has been changed to read: "This pattern is especially clear at Inyo Creek (Fig. 2a) and Providence Creek (Fig. 2b), which drain small, relatively undissected catchments. In particular, as can be seen in Fig. 2a by following a given elevation contour (black lines), travel distances (color bands) are longest in the valley axis and shortest at the ridges. Conversely, for a given travel distance (i.e. following a boundary between color bands), elevations are highest at the ridges and lowest in the valley axis."

**Comment 4:**

"In lines 166-167, it is said that "the joint distribution plots generally show dense concentrations of data points at low elevations for any given travel distance". This is not definitely true based on the color bar given in figure 4d. For instance in the Providence Creek, very small concentrations of data points exist at low elevations over a wide range of travel distances less than 4000 m. Similar observation can be made for the Inyo Creek for travel distances less than 3000 m, except some high concentration data points spanning around travel distances of 1500 m."

**Answer:**

The original text was unclear, thank you for pointing this out. The observation is about *relative* elevation for a given travel distance. In other words, looking only at a vertical column of data points, the thinnest concentrations occur at relatively high elevations (for that travel distance), and the thickest concentrations appear to occur at near the bottom of that vertical stack of points. However, the question of where the point densities are greatest is addressed in the next paragraph, so the best solution is to simply cut this sentence entirely, and better explain the point in the subsequent paragraph.

Changes in the manuscript:

The final sentence of the paragraph (which began at line 157) now reads "Meanwhile, many paired values are so common that they overlap, particularly along flowpaths that converge near the mainstem channel."

We have also modified the subsequent paragraph to better address this point, the relevant passage now reads: "These binned representations of the raw data show that, for a given travel distance, the lowest point densities (point area =  $100 \text{ m}^2$ ) generally occur at the highest relative elevations. As relative elevation decreases within a vertical stack of data, point density typically increases to a peak and then approaches zero at the channel elevation. In general, peak densities for a given travel distance occur closer to the channel than the ridge elevation, although there are notable exceptions."

**Comment 5:**

"In lines 169-171, it is said that "for a given travel distance, as elevation decreases, data point density generally increases to a peak and then quickly tapers to zero." Should not it be as elevation increases? At least all the plots (4c, 4e, 4f) show that for any given travel distance, the data point density goes to zero at the highest elevations (depicted by black colors)."

**Answer:**

This sentence was unclear. The changes made in response to Comment 4 also address the problems noted in this comment.

**Comment 6:**

"In lines 171-172, it is said that "they also show that the density of paired values is highest at 60 and 80% of the maximum travel distance". This is not true at all except for the Noyo River, while the statement is given in a general sense.

**Answer:**

Thank you again for highlighting the lack of clarity in this paragraph overall. In this sentence we wish to draw the reader's attention to where in each catchment the high point-densities are most common. We refer here to the bins colored red and brown, i.e. normalized density > 0.6. For Inyo and Providence Creeks, that occurs in the upstream-most third of the catchment; for Noyo it is the entire upper half of the catchment.

**Changes in the manuscript:**

The sentence beginning on line 171 has been rewritten to read: "Figure 4 (d-f) also shows that the greatest frequency of the high point-density (normalized density > 0.6) primarily occurs in the upper third of Inyo and Providence Creeks, and in the upper half of Noyo Creek."

**Comment 7:**

"First of all, direct comparison of figures 3 and 4 is not easy as the horizontal axes show the same quantity, but different ranges (authors might want to make it consistent throughout the paper). Second, figure 3 shows that at the Noyo River, the majority of the area pertains to long travel distances and low elevations. Can the authors explain why this is not reflected in figure 4f where highly dense data points correspond to mid travel distances and relatively high elevations?

**Answer:**

This is a very helpful comment in that it illuminates the challenges in thinking beyond the conventional use of hypsometry and width function, which represent elevation and travel distributions separately. In response to the first part of this comment, the horizontal axes in the six figure 4 panels are travel distance, scaled to extend over the same length on the page for each catchment. So in that respect they are directly comparable to the normalized travel distance shown on the horizontal axis in figure 3b. Figure 3a has normalized elevation on the horizontal axis, so comparison is a bit more difficult because in figure 4 elevation is shown on the vertical axes. But the scales of vertical exaggeration are adjusted to the ranges span the same length on the page, so if the reader can mentally rotate the figures the elevation axes are directly comparable.

What truly makes these two sets of figures difficult to compare is how area is represented. In both panels in figure 3, area is on the vertical axis, while in figure 4 area has no axis, rather it is distributed throughout the interior of the figure (and plotted as color contours in panels 4d-f). To see how the same data can produce these quite different graphical representations, one must mentally integrate across figure 4 to compare with figure 3. In response to the second part of this comment, in Figure 4f, when one integrates horizontally across a band of elevation, one sums area across nearly all travel distances for the relatively low elevations. This sum includes many if not most of the highest point density regions, and thus highest area regions. When one sums across the higher elevation bands, data are only encountered (and summed) at the highest travel distances and are thus sum to lower totals. This explains why the (hypsometric) distribution of area with elevation shown in figure 3a shows the majority of the area at relatively low elevations. There is no contradiction between the two figures, the underlying data are the same. The same exercise can be done in comparing figures 4f and 3b by integrating figure 4f vertically, thus summing area for a given travel distance. The vertical extent of the data cloud (the local relief for a given travel distance) gradually increases toward the right, as does the point density (and thus area), which is much greater on average for the longer travel distances. This pattern of vertical integration corresponds exactly to the distribution of area with travel distance shown in Figure 3b.

A key point is that plotting the joint distribution of elevation and travel distance reveals why the width function and hypsometric curves covary the way they do. It

shows where area is concentrated in the vertical and horizontal structure of the catchment. Another way to put it is Figure 3 can be derived from Figure 4, but not the other way around.

**Changes to the manuscript:**

We have added a new paragraph that will be inserted at line 173, which reads: "These patterns in the density of paired values of elevation and travel distance help explain the shapes of the corresponding hypsometry and width functions. For example, Figure 3 shows that in the Noyo Creek catchment the majority of area occurs at relatively long travel distances and relatively low elevations. Yet Figure 4f shows that this does not mean that the highest densities of catchment area occur at points that have both long travel distance and low elevation. Rather, low elevations dominate across all travel distances, and summing area horizontally across figure 4f leads to higher total area in the lower elevation bins of Figure 3a. Similarly, the Noyo catchment has greater relief at longer travel distances, and summing area vertically across fig. 4f leads to higher total area in the longer travel distance bins of Figure 3b. This comparison demonstrates that the joint distribution of elevation and travel distance reveals where area is distributed in the vertical and horizontal structure of the catchment in ways that the hypsometry and width function cannot."

We have also changed the first sentence of the next paragraph, to read: "Comparisons of the joint distributions between catchments also reveals significant differences that cannot be inferred from the conventional representations of vertical and horizontal catchment structure in Fig. 3."

**Comment 8:**

"In lines 231-233, it is said that "in landscapes where rates of precipitation and erosion are spatially variable and sometimes correlated, we expect the distributions of power and mean slopes to differ". Then the Inyo Creek catchment is mentioned as an example of this case. But comparing figures 5 and 6a does not support this at all, i.e., the spatial patterns of "water" power and mean slopes are very identical to each other in this catchment. How do the authors explain this?"

Answer: Thank you for pointing out the need for greater clarity in describing the differences in between the "distributions of power and mean slopes" at Inyo Creek (note that we did not use the phrase "spatial patterns"). The text beginning on line 237 is intended to describe the differences in question. Careful examination of the histograms in figures 5a and 6a (as well as 6b) shows that the shapes of the distributions are significantly different: for mean slope the distribution is negatively skewed, with a long tail of relatively low values, whereas the power distributions have a positive skew, with a long tail of relatively high values. As noted in the original text, the shape of the power contours are also shifted toward the ridges. These differences in contour pattern within one catchment are not as great as the differences between catchments shown in figure 5, however some of those differences can be attributed to the relative size of the catchments.

**Changes to the manuscript:**

We have changed the passage that began on line 237 to read: "When we combine these relationships for water and sediment production with the distribution of mean slopes using Equation 3, we can create histograms and maps showing the distributions of source-area power for the two materials, water and sediment (Fig. 6a-b). For both materials, the shape of the distributions shift from negative skew to positive skew, and the power contours are stretched towards the catchment divide, relative to the case of uniform precipitation and erosion (equivalent to Fig. 5a). The difference is greatest for the case of spatially varying erosion (Fig. 6b), due to the nonlinear relationship between erosion rate and elevation. Thus for catchments with spatial variation in the rate of production of water or sediment, mean slope distributions cannot reliably predict distributions of source-area power."

**Comment 9:**

"In figure 12, it seems that the hypsometric curves and width functions generated with the partially-synthetic formulation using actual profiles fits better to the real data than the fully-synthetic formulation using modeled profiles in the Noyo River. But the reverse is observed for the Providence Creek. Can the authors explain why the partially synthetic formulation using actual profiles should not always result in better fitting?"

**Answer:**

Thank you for this comment because it reveals that we mislabeled the model curves in Figure 12e, the hyposometry for the Noyo River catchment. The two model curves should be reversed. The reviewer is correct in assuming that the partially synthetic formulation using actual channel and ridge profiles should always result in a closer fit to the curves calculated from the DEM. This is the case for all six of the comparisons shown in Figure 12. To quantify the relative goodness of fit we have calculated the RMSE of the deviations between the model curves and the DEM. Comparing the partially synthetic with fully synthetic, for the Inyo Creek hypsometry and width function the values are 4.2 versus 5.7, and 4.6 and 5.0 respectively. Making the same comparison for Providence Creek, the values are 12.5 versus 15.7, and 12.3 and 14.2 respectively. And for Noyo River, the values are 2.1 versus 9.2, and 7.8 and 10.1 respectively.

**Change to the manuscript:**

We have revised Figure 12 to show the correct labeling of the curves for the Noyo River Hypsometry, and to list the RMSE values in the legends on each panel. The revised figure will be uploaded with this reply.

The caption to Figure 12 has been revised to include an additional sentence at the end: "Values in parenthesis indicate RMSE calculated by comparing model curves with DEM."

**Technical corrections:**

Line 438: figure 11 instead of 10 Line 444: figure 11A instead of 10A Line 456: figure 12 instead of 11 Line 477: figure 11 instead of 10 Line 478: figure 12 instead of 11 Line 482: figure 12a-b instead of 11a-b Line 484: figure 12c-d instead of 11c-d Line 486: figure 12c-d instead of 11c-d

**Response:**

Thank you for noting the incorrect numbering of the figures. The first problem is that figures 10 and 11 should be reversed. The fully synthetic model result should be figure 10 and the best fit beta distributions for Providence Creek and Noyo River should be figure 11.

**Changes to the manuscript:**

Figures 10 and 11 have been reversed in the sequence of figures. The figure numbering in the text has been double checked throughout the manuscript and all needed corrections have been made.

**Author Response to Comments by Referee #2**

We would first like the thank Reviewer 2 for their constructive comments. In this response we provide answers to each comment and detail the changes that will be applied in the revised manuscript. Please note that line numbers refer to the numbering of the original discussion manuscript.

**Overview**

"This paper proposes two new metrics to quantify landscape morphology based on the distribution of elevation and travel-distance, brought together in the concept of catchment power. Three examples of catchments with different morphologies are explored and a method is proposed by which artificial catchments with specified source-area power distributions can be synthesised. The paper's methods are certainly novel and raise some important questions about the formation of landscapes and the topic will be of broad interest to readers of ESurf. The manuscript is well written and carefully presented. I suggest that the manuscript is suitable for publication in ESurf, subject to satisfactory additional clarification and discussion of the following points."

**Response:**

Thank you for this positive summary and overall assessment of the paper. As detailed below we have done our best to provide the requested clarifications.

**Comment 1**:**

The paper concludes by stating that its major contribution is to offer a "fresh perspective". That's fine, but it would be better in my view to explain what new knowledge is available through the use of the new landscape metrics. The reader is left unclear on how this particular set of metrics might shed light on important problems in geomorphology.

**Answer:**

Thank you for highlighting the need for greater clarity on how these new metrics, source-area power and catchment power, might shed light on important problems in geomorphology. We have made changes (detailed below) the expand the discussion of future research opportunities and in the conclusion. In particular, we provide specific examples of questions for which these new metrics might help provide answers. These include what controls the size of sediments delivered to catchment outlets, and how does topography mediate the linkages between tectonics and climate?

**Changes in the manuscript:**

In the first paragraph of section 5.2 (future research opportunities), we have added text to help illustrate the claim that "this framework can be used to understand how the size distribution of sediments passing through a catchment outlet is influenced by weathering conditions at source elevations (Sklar et al., 2016), and by particle breakdown in transport (Attal and Lave, 2009)." The new text reads: "Specifically, the initial particle size produced on hillslopes may vary systematically with local climate, vegetation, and erosion rate, factors that commonly vary with elevation within catchments (Riebe et al., 2015). In the absence of particle size reduction in transport, the size distribution of sediments delivered to the outlet would then reflect the distribution of source elevations, weighted by the local erosion rate. Yet particle wear is likely to be significant except in small catchments underlain by exceptionally durable rock. The overall extent of particle size reduction in transport will depend on the distribution of travel distances and the rates of energy dissipation along those transport paths. Thus the evolution of sediment sizes in catchments, from source areas to the catchment outlet, and the resulting size distribution passing through the outlet, depend on the factors that together determine source-area power."

We have added a new paragraph to expand on the claim that "catchment power, the integral of source-area power for a given material over the entire catchment (equation 5), provides a metric for comparisons between catchments, and could be used to quantify, and help explain, the variation in topography across gradients in climate, tectonics and lithology." The new text reads "For example, Reiners et al., 2003, found a strong correlation between spatial variation in erosion rate and precipitation in the Cascade Mountains of Washington, but no corresponding trend in conventional topographic indices such as local relief. Catchment power, calculated for water delivered by precipitation, for sediment produced by erosion, or as the ratio of water to sediment power, could provide a metric that captures how

topography varies across gradients in precipitation and erosion. In this way, catchment power could help explain how topography mediates the linkage between climate and tectonics. Catchment power could also be used to compare numerical simulations of landscape evolution with real landscapes (Willgoose 1994; Willgoose et al., 2003), and contrast terrestrial catchments with catchments on Mars or Titan, where the topography reflects differing gravitational accelerations, fluids and rock properties (Mest et al., 2010; Burr et al., 2012)."

**Comment 2:**

The calculation of stream power (line 224) takes as the relevant slope the mean slope along the path to the catchment outlet. If the actual slope is close to the mean slope then this may be a good approximation. If not (for example, if the pathway might involve a very steep upper section with a long flat floodplain, or alternatively a high elevation plateau with a steep ravine descending from it) then the virtual velocity of sediment through the system will differ substantially, with important implications for residence time of sediment in floodplains etc (which is itself relevant geochemical residence times in the catchment, cosmogenic methods, and carbon sequestration). This warrants some further discussion.

**Answer:**

This is a very helpful comment in that it highlights the need to explain how sourcearea power is different from stream power. There are two key differences. First stream power uses the entire upstream contributing area to calculate the material flux, whereas the contributing area for source-area power is limited to the smallest unit of analysis, such as a single pixel in a DEM. Second, stream power quantifies the local rate of energy dissipation across a short distance, such as a reach of river represented by the distance between two pixels, whereas source-area power averages energy dissipation over the entire travel distance from source to catchment outlet. Unlike stream power, source-area power quantifies the production rate of material potential energy in terms of the position of the source location relative to the catchment outlet. This provides a distinct metric for analyzing spatial patterns in how energy is produced within catchments, relative to the distance over which the effects of energy dissipation are realized.

**Changes in the manuscript:**

A new paragraph has been inserted following the paragraph containing equation 3 at line 224. "Source-area power is distinct from stream power, which is how energy dissipation in landscapes is commonly quantified (Rodriguez-Itrube et al., 1992; Lague, 2014). Stream power uses the entire upstream contributing area to calculate the material flux, whereas the contributing area for source-area power is limited to the smallest unit of analysis, such as a pixel in a DEM. Moreover, stream power quantifies the local rate of energy dissipation across a short distance, such as a reach of river represented by the distance between two pixels, whereas source-area power averages energy dissipation over the entire travel distance from source to catchment outlet. Hence, unlike stream power, source-area power quantifies the production rate of material potential energy in terms of the position of the source location relative to the catchment outlet. This provides a distinct metric for analyzing spatial patterns in how energy is produced and dissipated within catchments."

**Comment 3:**

In section 3 (line 243 onwards) the notation switches from the generic subscripts i,j to w and s for water and sediment, and the dimensionless ratio \omega^\* is defined as the ratio of source-area power of water per mass of sediment. The intuitive/conceptual significant of this ratio is not clear, which makes it hard to interpret the values 36–653 in the subsequent paragraph.

**Answer:**

We agree that the motivation for this analysis was poorly articulated in the original draft. The goal of comparing source-area power for water with sediment production rate is to explore how the topography, as expressed in the joint distribution of elevation and travel distance, reflects the spatial variation and relative importance of water-mediated sediment transport processes, such as overland, debris, and fluvial flows, as opposed to primarily gravity-driven processes such as creep and landslides. We have added several sentences to the paragraph beginning at line 243 to more clearly motivate this analysis.

Thank you for pointing out the inconsistency in the sub-script notation. The first subscript should always refer to a location and the second subscript should refer to a material, or in this case a ratio of one material to another. We have adjusted the notation for the quantity defined in equation 4 to be consistent with this subscript convention.

**Changes in the manuscript:**

The new text reads "Comparisons of source-area power and production rates for different materials may provide insight into the spatial variation of catchment processes. For example, sediment produced by erosion at source areas is transported to the outlet by a combination of primarily gravity-driven processes, including creep and landslides, and by water-mediated processes such as overland, debris, and fluvial flows. Catchment topography, as expressed in the joint distribution of elevation and travel distance, may reflect the spatial variation and relative importance of these different processes."

The symbol for the dimensionless ratio of water source-area power to sediment mass production rate is now written as  $\omega_{i,w/s}^*$  in equation 4 and in the accompanying text.

**Specific comments / Minor points**

Line 86 Tarbotton -> Tarboton

Response: Thank you for catching this, the misspelling has been corrected.

**Catchment power and the joint distribution of elevation and travel distance to the outlet.**

Leonard S. Sklar1, Clifford S. Riebe2, Claire E. Lukens2, Dino Bellugi3 3 4 1Department of Earth and Climate Sciences, San Francisco State University, San Francisco, CA 94132 5 USA 2Department of Geology and Geophysics, University of Wyoming, Laramie, WY 82071 USA 6 7 3Department of Earth, Atmospheric and Planetary Sciences, MIT, Cambridge, MA 02139 USA 8 Correspondence to L.S. Sklar (leonard@sfsu.edu) 9 Abstract The delivery of water, sediment and solutes by catchments is influenced by the distribution of 10 source elevations and their travel distances to the outlet. For example, elevation affects the magnitude and 11 phase of precipitation, as well as the climatic factors that govern rock weathering, which influence the 12 production rate and initial particle size of sediments. Travel distance, in turn, affects the timing of flood 13 peaks at the outlet and the degree of sediment size reduction by wear, which affects particle size 14 distributions at the outlet. The distributions of elevation and travel distance have been studied extensively 15 but separately, as the hypsometric curve and width function. Yet a catchment can be considered as a 16 collection of points, each with paired values of elevation and travel distance. For every point, the ratio of 17 elevation to travel distance defines the mean slope for transport of mass to the outlet. Recognizing that 18 mean slope is proportional to the average rate of loss of potential energy by water and sediment during 19 transport to the outlet, we use the joint distribution of elevation and travel distance to define two new 20 metrics for catchment geometry: "source-area power," and the corresponding catchment-wide integral 21 "catchment power." We explore patterns in source-area and catchment power across three study catchments 22 spanning a range of relief and drainage area. We then develop an empirical algorithm for generating 23 synthetic source-area power distributions, which can be parameterized with data from natural catchments, 24 This new way of quantifying the three-dimensional geometry of catchments can be used to explore the 25 effects of topography on the distribution on fluxes of water, sediment, isotopes and other landscape 26 products passing through catchment outlets. and may provide a fresh perspective on problems of both 27 practical and theoretical interest.

**28 1. Introduction**

- 29 The physical and ecological dynamics of rivers are influenced by upstream sources of water,
- 30 solutes, and sediment. These materials are produced at rates that vary from source to source depending on
- 31 factors such as precipitation, weathering, erosion, and ecosystem productivity. Spatial variations in these
- 32 factors commonly correspond to differences in elevation. For example, elevation influences both the
- 33 magnitude and phase of precipitation (Roe, 2005; Minder et al., 2011), the climatic factors that govern rock
- 34 weathering (White and Blum, 1995; Riebe et al., 2004), the particle size and production rate of sediment
- 35 from slopes (Marshall and Sklar, 2012; Riebe et al., 2015; Sklar et al., 2016), and both the distribution of

1

**Leonard Sklar 8/5/2016 12:12 PM**

**Deleted: &**

Leonard Sklar 5/25/2016 10:13 AM Deleted: , Leonard Sklar 5/25/2016 10:10 AM Deleted: and Leonard Sklar 5/25/2016 10:10 AM Deleted: . Leonard Sklar 5/25/2016 10:10 AM Deleted: 
[revised manuscript text omitted]

7

**Leonard Sklar 7/16/2016 5:36 PM Deleted: integrates**

Leonard Sklar 7/16/2016 5:38 PM **Deleted:** on a hillslope

Leonard Sklar 7/16/2016 5:39 PM Deleted: i.e

(2).

(3).

Leonard Sklar 7/16/2016 5:39 PM Formatted: Font:Not Italic

Leonard Sklar 7/16/2016 9:21 AM Formatted: Line spacing: 1.5 lines 292 commonly quantified (Rodriguez-Itrube et al., 1992; Lague, 2014). Stream power uses the entire upstream 293 contributing area to calculate the material flux, whereas the contributing area for source-area power is 294 limited to the smallest unit of analysis, such as a single pixel in a DEM. Moreover, stream power 295 quantifies the local rate of energy dissipation across a short distance, such as a reach of river represented by 296 the distance between two pixels, whereas source-area power averages energy dissipation over the entire 297 travel distance from source to catchment outlet. Hence, unlike stream power, source-area power quantifies 298 the production rate of material potential energy in terms of the position of the source location relative to the 299 catchment outlet. This provides a distinct metric for analyzing spatial patterns in how energy is produced 300 and dissipated within catchments. 301 The concept of source-area power allows us to explore the possible implications of variability in 302 the ratio of elevation to travel distance (i.e., the mean slope) on the production and delivery of water, 303 solutes, and sediment across catchments. For example, in landscapes where the rate of precipitation or 304 erosion is spatially uniform, we expect the distribution of source-area power for the water or sediment to be 305 identical to the distribution of the mean slopes of source areas. In contrast, in landscapes where rates of 306 precipitation and erosion are spatially variable and sometimes correlated (Reiners et al., 2003;, Burbank et 307 al. 2003), we expect the distributions of power and mean slopes to differ. This is the case at Inyo Creek 308 where mean annual precipitation increases with elevation from 290 mm yr-1 at the outlet to 710 mm yr-1 at 309 the catchment divide (Prism Climate Group, 2014), and the rate of production of sediment by erosion has 310 been estimated to increase exponentially with elevation from 0.03 mm yr-1 at the outlet to 1.5 mm yr-1 at the 311 divide (Riebe et al., 2015). When we combine these relationships for water and sediment production with 312 the distribution of mean slopes using Equation 3, we can create histograms and maps showing the 313 distributions of source-area power for the two materials, water and sediment (Fig. 6a-b). For both materials, 314 the shape of the distributions shift from negative skew to positive skew, and the power contours are 315 stretched towards the catchment divide, relative to the case of uniform precipitation and erosion (equivalent 316 to Fig. 5a). The difference is greatest for the case of spatially varying erosion (Fig. 6b), due to the 317 nonlinear relationship between erosion rate and elevation. Thus for catchments with spatial variation in the 318 rate of production of water or sediment, mean slope distributions cannot reliably predict distributions of 319 source-area power. 320 Comparisons of source-area power and production rates for different materials may provide 321 insight into the spatial variation of catchment processes. For example, sediment produced by erosion at 322 source areas is transported to the outlet by a combination of primarily gravity-driven processes, including 323 creep and landslides, and by water-mediated processes such as overland, debris, and fluvial flows. 324 Catchment topography, as expressed in the joint distribution of elevation and travel distance, may reflect 325 the spatial variation and relative importance of these different processes. Because the altitudinal gradients 326 in erosion and precipitation at Inyo Creek are known, we can use them to explore how the source-area 327 power of water, relative to the amount of sediment that must be produced on hillslopes and transported to

Leonard Sklar 7/16/2016 9:20 AM Formatted: Font:10 pt

Leonard Sklar 7/16/2016 9:20 AM Formatted: Font:10 pt Leonard Sklar 7/16/2016 9:20 AM Formatted: Font:10 pt Leonard Sklar 7/16/2016 9:20 AM Formatted: Font:10 pt

Leonard Sklar 7/16/2016 9:20 AM Formatted: Font:10 pt Leonard Sklar 7/16/2016 9:20 AM Formatted: Font:10 pt Leonard Sklar 7/16/2016 9:10 AM Formatted: Indent: First line: 0" Leonard Sklar 7/16/2016 9:10 AM Deleted:

Leonard Sklar 5/27/2016 11:08 AM Deleted: arrive at Leonard Sklar 5/27/2016 11:09 AM Deleted: spatial Leonard Sklar 5/28/2016 2:04 PM Deleted: In both cases Leonard Sklar 5/27/2016 11:09 AM Deleted: , especially Leonard Sklar 5/27/2016 11:10 AM Deleted: in

Leonard Sklar 7/14/2016 7:49 PM Deleted: B

Leonard Sklar 5/28/2016 2:06 PM Deleted: varies across the catchment

[revised manuscript text omitted]